**Subject Area:**
cellular biology/molecular biology

TFIIH, cancer, transcription, gene expression, RNA polymerase II

**Author for correspondence:**
Mario Zurita
e-mail: marioz@ibt.unam.mx

# Disruption of TFIIH activities generates a stress gene expression response and reveals possible new targets against cancer

Maritere Uriostegui-Arcos[1], Rodrigo Aguayo-Ortiz[3,4],
María del Pilar Valencia-Morales[1], Erika Melchy-Pérez[2], Yvonne Rosenstein[2],
Laura Dominguez[3] and Mario Zurita[1]

[1]Departamento de Genética del Desarrollo y Fisiología Molecular, Instituto de Biotecnología, and
[2]Departamento de Biomedicina Molecular y Bioprocesos, Instituto de Biotecnología, Universidad Nacional Autónoma de México, Cuernavaca Morelos 62250, Mexico
[3]Departamento de Fisicoquímica, Facultad de Química, Universidad Nacional Autónoma de México, Mexico City 04510, Mexico
[4]Center for Arrhythmia Research, Department of Internal Medicine, Division of Cardiovascular Medicine, University of Michigan, Ann Arbor, MI 48109, USA

MZ, 0000-0002-8404-2173

Disruption of the enzymatic activities of the transcription factor TFIIH by the small molecules Triptolide (TPL) or THZ1 could be used against cancer. Here, we used the MCF10A-ErSrc oncogenesis model to compare the effect of TFIIH inhibitors between transformed cells and their progenitors. We report that tumour cells exhibited highly increased sensitivity to TPL or THZ1 and that the combination of both had a synergic effect. TPL affects the interaction between XPB and p52, causing a reduction in the levels of XPB, p52 and p8, but not other TFIIH subunits. RNA-Seq and RNAPII-ChIP-Seq experiments showed that although the levels of many transcripts were reduced, the levels of a significant number were increased after TPL treatment, with maintained or increased RNAPII promoter occupancy. A significant number of these genes encode for factors that have been related to tumour growth and metastasis, suggesting that transformed cells might rapidly develop resistance to TPL/THZ inhibitors. Some of these genes were also overexpressed in response to THZ1, of which depletion enhances the toxicity of TPL, and are possible new targets against cancer.

## 1. Introduction

Cancer cells are known to be addicted to high levels of transcription as the enhanced expression of a plethora of molecules is required for the generation and the maintenance of a transformed phenotype [1,2]. This information suggests that the different factors that participate in general transcriptional activation by RNA polymerase II (RNAPII) could be targets for treating cancer. Since RNAPII is not able to recognize the promoter and initiate transcription in a regulated way by itself, it requires the assembly of what is known as the pre-initiation complex (PIC) at promoters. Generally, the PIC includes the TFIID complex, RNAPII, TFIIB, TFIIA, TFIIF, TFIIE and TFIIH [3]. In metazoans, during transcriptional activation, RNAPII synthesizes a transcript with a length of 20–120 nucleotides and then it pauses [4,5]. The release of paused RNAPII is conducted by the positive-elongation factor p-TEFb [6,7].

A component of the PIC and an interesting target to affect transcription—and therefore cancer—is TFIIH [8,9]. TFIIH is a complex of 10 subunits composed of the CAK subcomplex containing CDK7, CYCH and MAT1, which also

participates in cell cycle control, and the core subcomplex that is part of the mechanism of nucleotide excision repair (NER) [10]. The core subcomplex is composed of p8, p34, p44, p52, p62, XPB and XPD subunits (the last two are DNA helicases/ATPases) [11]. Together, the CAK and the core form the 10-subunit TFIIH complex, which participates in transcription [11,12]. The role of TFIIH in RNAPII transcription involves phosphorylation of Ser 5 in the RNAPII CTD ($p^{Ser5}$CTD RNAPII) by CDK7, which is important for transcription initiation, recruitment of the CAP enzyme, other modifications and mRNA processing [12–14]. By contrast, XPB functions as an ATP-dependent translocase that rotates DNA to open it around the transcription initiation site, facilitating the synthesis of RNA by RNAPII [15,16]. Thus, compounds that affect the TFIIH functions have been found or developed as strong candidates to combat cancer. For instance, the chemical compound THZ1 (phenylaminopyrimidine) and related substances inhibit the kinase activity of CDK7 by binding a protein region outside of its catalytic domain (Cys312) [17]. Although THZ1 is very effective in killing different types of cancer cells, it has not yet been clinically tested in cancer patients [18]. On the other hand, the diterpenoid epoxide, triptolide (TPL), inhibits the ATPase activity of XPB by covalently binding its catalytic domain [19]. TPL and its derivatives have been shown to kill different types of cancer cells and minelide, a triptolide derivative is currently in clinical trials for the treatment of pancreatic cancer [20,21]. In addition, TPL has been used as a tool to study transcription initiation and promoter-proximal pausing [22–24]. Although many studies demonstrate the potential use of TPL against cancer [25–27] and indicate how this drug affects transcription initiation, studies of how TPL affects global gene expression between cancer cells and their progenitors are still needed. Additionally, it is not known whether TPL and THZ1 cause a similar effect in cells or whether TPL affects TFIIH integrity. In addition, transcriptional response studies are still limited or analysed with brief incubation times and not when the effect of TPL on cell homeostasis is initiating. In this work, we addressed these points using an inducible oncogenesis model.

## 2. Results

### 2.1. Triptolide and THZ1 preferentially kill transformed cells, and the combination has a synergic effect

To study the effects of TPL and THZ1 on cancer cells and their progenitors, we used the MCF10A-ErSrc cell line as an oncogenesis model [28]. After 36–72 h of incubation with tamoxifen, which activates v-Src oncoprotein, the MCF10A-ErSrc line achieves multiple features associated with cellular transformation like: high proliferation, lost adherence, mammospheres formation and generates tumours metastasis in immunocompromised mice. In this study, cells with these characteristics are referred to as tamoxifen-treated (TAM) and their progenitors as non-treated (NT). Since phosphorylated STAT3 (pSTAT3) is required for cellular transformation through IL6 and NF-kB, thereby it was used as a transformation control in this cell line [28,29].

To analyse whether NT cells are more sensitive than TAM cells when TFIIH is affected, we evaluate by flow cytometry viability of NT and TAM cells which were incubated with TPL, THZ1 or both chemicals at different concentrations and for different times (figure 1a). The viability of both NT and TAM cells was highly affected by the presence of TPL, with TAM cells being more sensitive (figure 1a). Following incubation with 100 nM TPL for 72 h, approximately 90% of TAM cells died (figure 1a). However, at the same concentration and incubation time, more than 50% of NT cells were still viable. By contrast, at a concentration of 250 nM THZ1 for 72 h, approximately 64% of TAM cells died and only 36% of NT cells died (figure 1a). Interestingly, simultaneous incubation of NT and TAM cells with both chemicals had an additive effect on cell viability (figure 1a), as practically all TAM cells died after 48 h when incubated with TPL (100 nM) and THZ1 (250 nM) (figure 1a). However, under those conditions and even after 72 h, approximately 40% of NT cells remained viable (figure 1a). Thus, the combination of TPL and THZ1 is better than the single treatment with each of those drugs and preferentially target TAM cells for cell death via apoptosis (electronic supplementary material, figure S1a–d). Also, drug combination analysis by the Chou–Talalay method [30] showed combination index (CI) values < 1.0 in TAM cells, which denotes a synergistic pharmacological effect (electronic supplementary material, figure S1e). Interestingly, TPL/THZ1 treatment in NT cells has a synergistic effect only in the highest concentrations (electronic supplementary material, figure S1e).

Next, we evaluated by flow cytometry assays the effect of TPL, THZ1 and the combination of both at different concentrations and times on proliferation and cell cycle progression in NT and TAM cells (figure 1b,c). Figure 1b shows that after 72 h of incubation with 25 nM TPL, TAM cells stopped after two cycles of proliferation; and that NT cells required 100 nM TPL to stop proliferating (figure 1b). Similarly, TAM and NT cells stopped proliferating, when incubated with 100 nM or 250 nM THZ1 for 72 h, respectively (figure 1b). Interestingly, when incubated with both TFIIH inhibitors NT and TAM cells stopped proliferating with only 25 nM TPL and 50 nM THZ1, confirming the synergic effect of these drugs (figure 1b). Furthermore, we found that in the presence of TPL and THZ1, cells were arrested at the $G_1$ phase and that lower concentrations of TPL and THZ1 were needed for TAM cells (figure 1c). Taken together, these results indicate that TAM cells are more sensitive to TPL and THZ1 than their NT cells counterparts. TAM cells stopped proliferating and were arrested at $G_1$ at lower concentrations and shorter incubation times when incubated with either drug. Importantly, simultaneous treatment with TPL and THZ1 had a significantly more severe effect on TAM cells than on NT cells as well as either drug used independently, underscoring the potential of simultaneously inhibiting different TFIIH activities with TPL and THZ to develop alternative therapies for cancer treatment.

### 2.2. TPL interferes with the XPB–p52 interaction, inducing the degradation of the XPB–p52–p8 submodule of TFIIH

Based on our previous analysis of TFIIH mutants in *Drosophila* [31], we sought to explore the effect of TPL on the XPB levels in NT and TAM cells. Cells were incubated with 125 nM TPL for different times. Since the disruption of transcription by RNAPII causes degradation of this enzyme, we also evaluated levels of RNAPII as well as of other TFIIH subunits. As expected, levels of RNAPII—and therefore $p^{ser5}$CTD

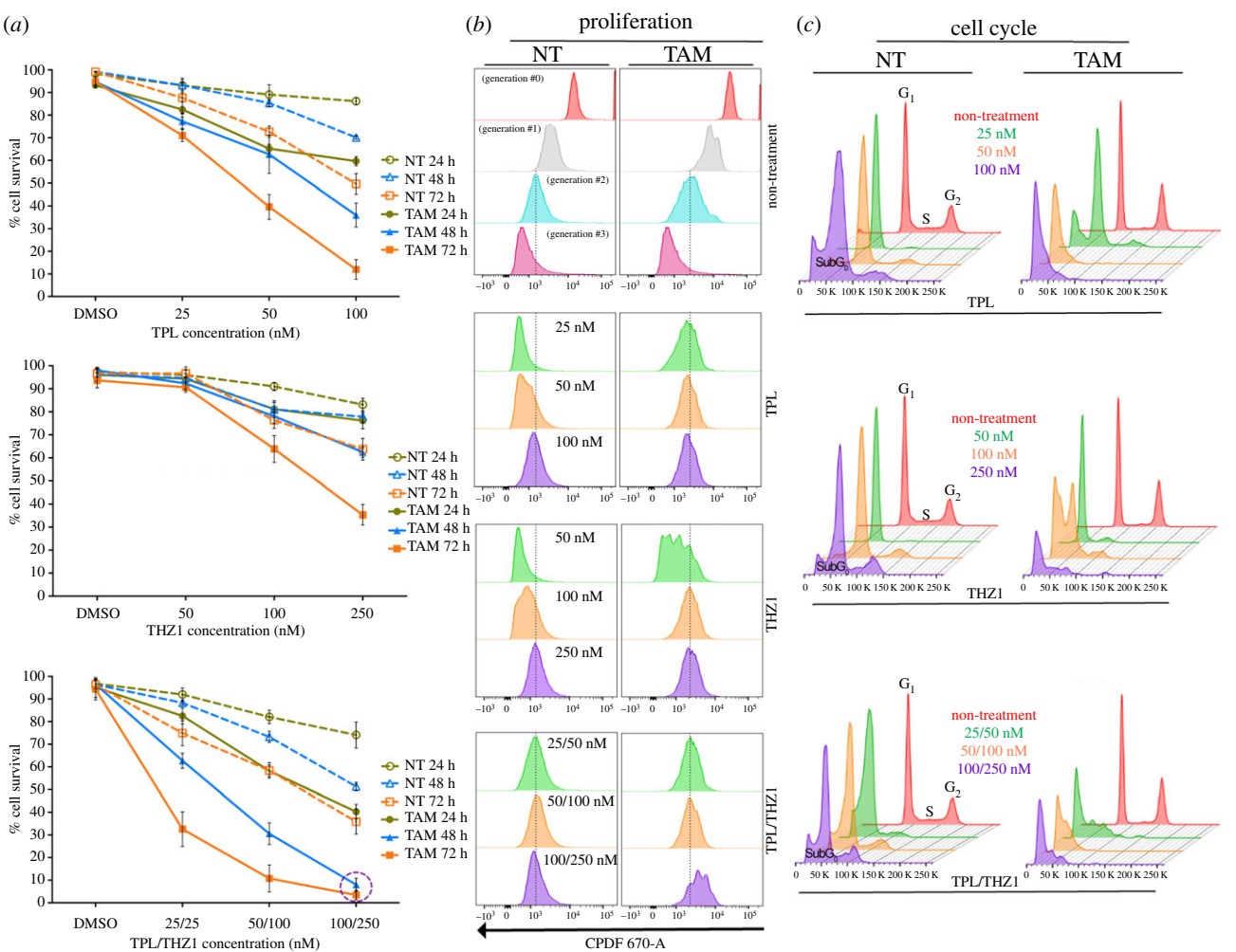

**Figure 1.** Triptolide (TPL) and THZ1 preferentially kill transformed cells and the combinatory of both substances potentiate their effect. Flow cytometry assays to determine cellular viability, proliferation and cell cycle arrest in tamoxifen-treated (TAM) and non-treated (NT) cells. (a) Cellular viability observed in cells incubated with TPL (upper panel), THZ1 (middle panel) and the combination of THZ1 and TPL (lower panel). The concentrations and incubation time used are indicated in the figure. Data are presented as mean values ± s.d. (standard deviation) from three independent experiments. (b) Proliferation assay in TAM and NT cells incubated with TPL (upper panel), THZ1 (middle panel) and the combination of THZ1 and TPL (lower panel). The first panel shows non-treatment cells throughout its generational duplications. (c) Cell cycle arrest assay in TAM and NT cells incubated with TPL (upper panel), THZ1 (middle panel) and the combination of THZ1 and TPL (lower panel). Graphs in (b) and (c) show concentrations used for 72 h of treatment and correspond a representative example from three biological replicates.

RNAPII—decreased as a result of incubating the cells with TPL (figure 2a). A clear reduction in XPB protein was also observed, which was greater in TAM cells (figure 2a). Furthermore, levels of the p52 and p8 subunits were also diminished in response to TPL exposure (figure 2a). However, levels of other subunits of TFIIH, such as XPD, p62, CDK7, CYCH and MAT1, were not affected (figure 2a; electronic supplementary material, figure S2a). As expected, incubation with THZ1 for different times reduced the levels of $p^{ser5}$CTD RNAPII, but it did not affect the levels of this enzyme or the levels of any TFIIH subunit, including the CAK subcomplex, consistent with previous reports [32] (electronic supplementary material, figure S2a).

p52 and p8 subunits have direct contact with XPB and modulate its ATPase activity [31,33]. Since our results suggested that the binding of TPL to the ATPase domain of XPB destabilizes XPB as well as p52 and p8, we investigated whether TPL causes a distortion of XPB that limits the interaction of XPB with p52 and p8. To achieve this aim, we used the public information recently reported for the structure of the human TFIIH core by cryo-electron microscopy [34]. TPL inhibits XPB-ATPase function through the formation of a covalent bond between the C12 carbon (12,13-epoxide group) on the inhibitor and the sulfur atom of the Cys342

residue of XPB ($TPL_{C12}$-Cys342) [35]. The isolated XPB–p52–p8 putative submodule was employed to perform molecular dynamics (MD) simulations of the covalent docking of the optimized TPL structure to the Cys342 residue of XPB (figure 2b). Our covalent docking study showed that the TPL binding site (TBS) in XPB is located at the interface of the helicase domains (HD1 and HD2), which are mainly constituted by DEVH box residues (electronic supplementary material, figure S3). During the MD simulations, a per-residue contact analysis shows that the presence of TPL at the HD1–HD2 interface altered the number of contacts between both domains (electronic supplementary material, figures S3d and S3g), which may lead the separation of the domains and that the dissociation could be due to allosteric modulation guided by the loss of interactions between the XPB N-terminal domain (NTD) and p52 and between XPB HD2 and p52/p8.

To confirm the MD results we performed split-GFP-complementation assays between XPB and p52 by using the tripartite split-GFP association system [36], which is based on a tripartite association between two GFP tags (20 amino acids long each) fused to interacting protein partners, and the complementary GFP1-9 detect. Stable HEK-293 cells expressing GFP1–9 were co-transfected with GFP10-P52 and XPB-GFP11 constructs an

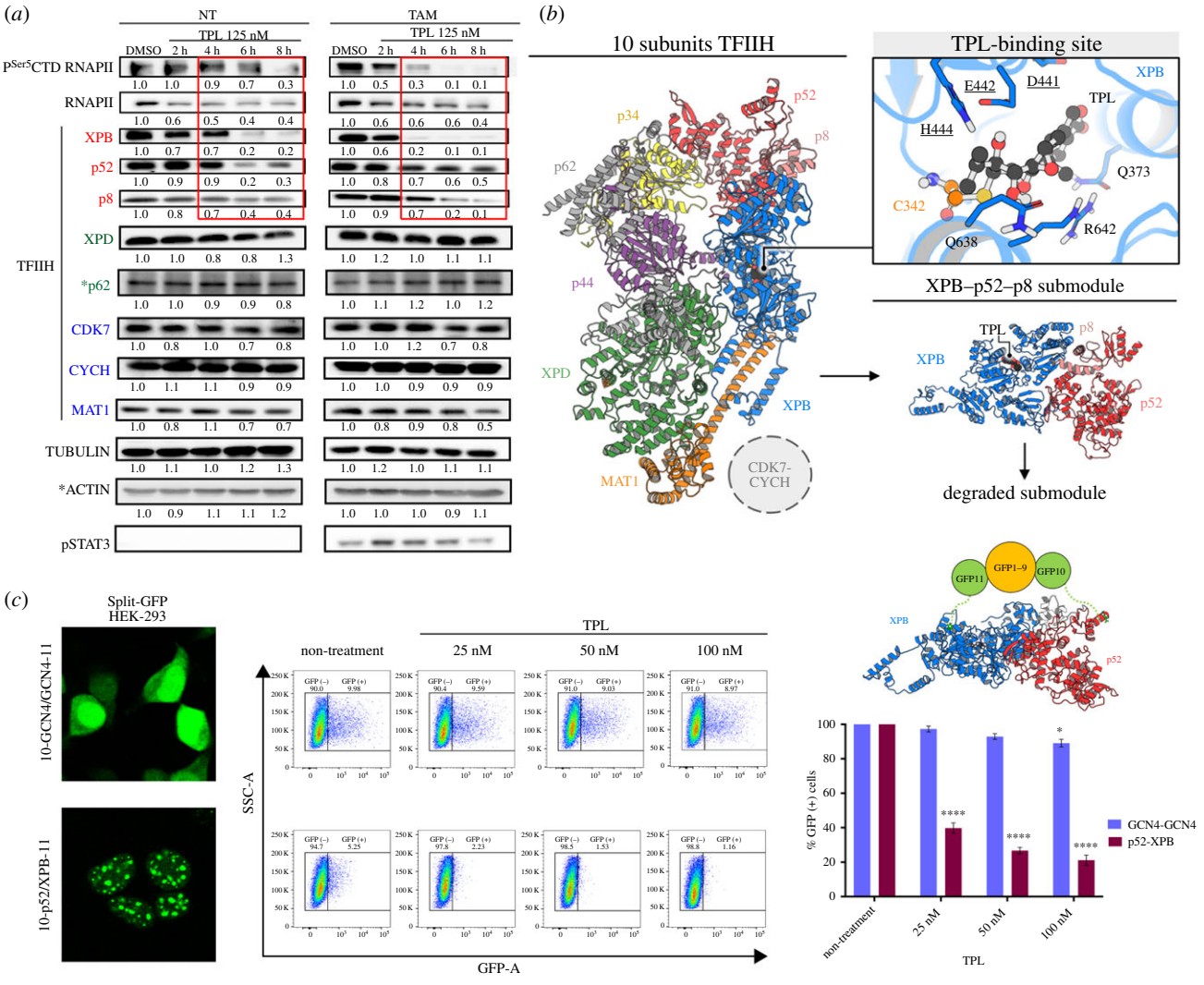

**Figure 2.** TPL induces the degradation of the XPB–p52–p8 submodule of TFIIH. (*a*) Western blots showing RNAPII CTD and the $p^{Ser5}$CTD RNAPII, and some TFIIH subunits (XPB, p52, p8, XPD, p62, CDK7, CYCH and MAT1) from cells incubated with TPL 125 nM for 2, 4, 6, 8 h in comparison to the control with DMSO for 8 h. (*) Independent western blots using the same extracts. Note that the TFIIH subunits, XPB, p52 and p8 protein levels decrease as the incubation time with TPL advances (box in red). Densitometric analyses were performed using tubulin and actin (only for p62) as loading control; the relative quantification is indicated under each blot and the data showed as a representative example from three biological replicates. The p-STAT3 is used as a transformation control. (*b*) Computational molecular dynamics model that proposes the mechanism of the XPB–p52–p8 submodule dissociation from the TFIIH complex and degradation due to the covalent binding of TPL to XPB. The figure on the left shows the three-dimensional structure of TFIIH coloured by its components: XPB (blue), XPD (green), p8 (pink), p34 (yellow), p44 (purple), p52 (red), p62 (grey) MAT1 (orange). The upper right figure depicts TPL (black) binding mode in XPB. TPL binding site (TBS) residues are shown in blue, underlining the DEVH box amino acids, and C342 forming the covalent bond with TPL is highlighted in orange. (*c*) TPL interferes with the p52–XPB interaction. A representative image from at least three independent Split-GFP complementation assays between the p52 and XPB subunits expressed in HEK-293 cells is shown in the left panel. The middle panel shows an example of the cytometry measure of the GFP fluorescent activity by the cell in the GCN4-GCN4 split-GFP homodimer complementation used as control and the p52–GFP–XPB split complementation. Note that at 25 nM of TPL incubation by 28 h practically no fluorescent cells in the p52–XPB construct are detected. The right panel shows a kinetic assay of three independent experiments where the percentage of live and positive cells for GFP is shown. Significant differences were analysed by Two-way ANOVA with corrections for multiple comparisons, always comparing with the non-treated column. Statistical significance is indicative *$p < 0.05$, ****$p < 0.0001$. The structure at the top of the histogram depicts the p52–GFP–XPB complementation, showing that the localization of the GFPs is compatible with the formation of a functional TFIIH complex with a reasonable distance between fused GFP10 and GFP11 fragments.

incubated with TPL at different low concentrations for 28 h (figure 2*c*). As a control, we used a GCN4 homodimeric interaction (GFP10-GCN4 and GCN4-GFP11) previously reported [36]. A three-dimensional representation of the XPB–GFP–p52 complex shows that the localization of GFP is compatible with the interaction between XPB and p52 (figure 2*c*). The GFP fluorescence signal was quantified only in living cells by flow cytometry. After the TPL treatment a clear reduction in the fluorescence is observed in the p52–GFP–XPB-complementation cells, but not in the control cells (figure 2*c*). These *in vivo* results are in agree with structural modelling results that suggest that TPL interferes with the binging between XPB

and p52. Altogether, the results of this section suggest that XPB, p52 and p8 form a submodule in the core of TFIIH and that TPL besides inhibiting the XPB-ATPase activity, also cause the XPB–p52–p8 destabilization without affecting the rest of the TFIIH subunits.

## 2.3. Analysis of the transcriptome of TPL-treated cells shows an unexpected gene expression response

While analysing the transcriptome of TFIIH mutants in *Drosophila*, we previously reported that not all genes responded

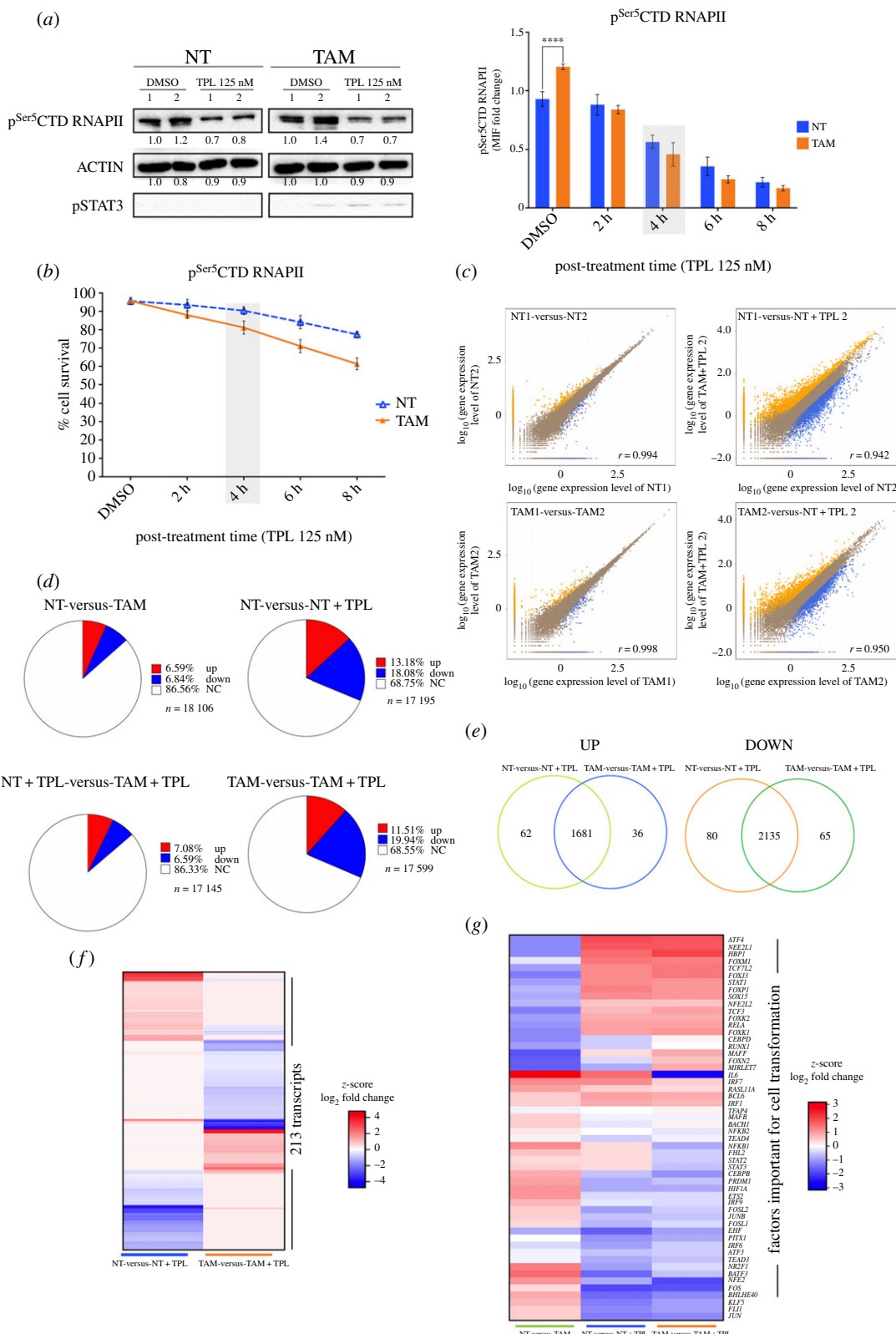

**Figure 3.** (Caption overleaf.)

equally and that the transcript levels of numerous genes increase in TFIIH mutant tissues [37,38]. Therefore, to explore whether TPL also generates a differential effect on gene expression in TAM and NT cells, we analysed the transcriptomes of these cells after incubation with 125 nM TPL for 4 h, a concentration found to reduce equally RNAPII levels by half in NT and TAM cells, to affect mildly cell viability (figure 3a,b).

Approximately 18 500 different transcripts were identified in both TAM and NT cells (electronic supplementary material,

figure S4a). A Pearson's correlation analysis between NT and TAM cells with and without TPL showed, as expected, that the treatment with TPL caused a reduction in the transcript levels of some genes, but intriguingly, the levels of other transcripts increased (figure 3c; electronic supplementary material, figure S4b). Induction of the transformed phenotype of MCF10A-ErSrc cells reduced the expression of 6.84% and increased the expression of 6.59% of the genes (figure 3d). When we compared the effect of TPL in NT and TAM cells, the RNA levels of approximately 68% of the genes did not

**Figure 3.** (*Overleaf.*) Transcriptome analysis of TPL-treated cells. (*a*) Left panel show western blots (WB) to evaluate the p$^{Ser5}$CTD RNAPII in tamoxifen <u>non-treated</u> (NT) and <u>tamoxifen</u>-treated (TAM) cells incubated with triptolide (TPL) for 4 h at 125 nM or DMSO for 4 h as a control. Densitometric analyses were performed using actin as a loading control. The relative quantification is indicated under each blot from two biological replicates. pSTAT3 is used as a transformation control. The cells were collected and divided into two fractions, one for WB and the other for RNA-Seq analysis. The right panel indicates the quantification of p$^{Ser5}$CTD RNAPII by flow cytometry, NT (blue) and TAM cells (orange), the grey box, indicates the time that p$^{Ser5}$CTD RNAPII has decreased approximately 50%. (*b*) Cell viability of the NT (blue) and TAM cells (orange) incubated with 125 nM of TPL for 2, 4, 6 and 8 h. DMSO (incubated for 8 h) was used as control. It can be appreciated that at 4 h (grey box) the cell viability is not very compromised. (*c*) Pearson's correlation between two of the replicas of the RNA-Seq data (left) and between two samples (right); the *x*- and *y*-axes represent the log$_{10}$ value of gene expression. The blue dots represent the downregulation transcripts; orange dots upregulation transcripts and brown dots non-affected. (*d*) Percentages of up and downregulated genes in NT, NT cells incubated with triptolide for 4 h at 125 nM (NT + TPL), TAM cells and TAM cells incubated with triptolide for 4 h at 125 nM (TAM + TPL). In red are represented the transcript percentages that increased, in blue the transcript percentages that are down and in white the transcripts with no significant log$_2$ fold change. (*e*) Venn diagram showing the percentage of up and down, unique and common transcripts between NT and TAM cells treated TPL. (*f*) Heat map comparing 213 transcripts that are differentially expressed between NT and TAM cells after TPL treatment. (*g*) Heat map showing the response to TPL in the expression of genes that are either up or downregulated in the establishment of the transformed phenotype. Note that TPL induces the increase in expression of most of the genes downregulated for the transformed phenotype and reduces the expression of most of the genes upregulated to maintain the transformed phenotype. Right panel (*a*) and graph (*b*) represent three biological replicates while left panel (*a*) and graphs (*c*) to (*g*) summarizing the results of two biological replicates. In graph (*a*-right) and (*b*) data are mean ± s.d. (standard deviation). Significant differences were analysed by Two-way ANOVA with corrections for multiple comparisons. Statistical significance is indicated (****$p < 0.0001$).

---

significantly change, probably because, in the conditions used in this experiment, the reduction in transcription initiation of many genes is not detected by RNA-Seq (figure 3*d*). However, in both, NT and TAM cells, approximately 11% of the gene transcript levels were increased, and approximately 19% were decreased (figure 3*d*). Among the downregulated genes, 2135 transcripts were shared between NT and TAM cells. Among the upregulated genes, 1681 transcripts were common to NT and TAM cells, 62 were exclusive to NT cells and 36 were exclusive to TAM cells (figure 3*e*; electronic supplementary material, table S1). The expressions of 213 genes were exclusive to either NT or TAM cells (figure 3*f*). Some genes as *SOX9*, *RETL1* and *IGFBP3*, were downregulated in NT cells and upregulated in TAM cells in response to TPL. However, we also detected genes whose transcripts were upregulated in NT cells and downregulated in TAM cells, such as *FAM111B* and *F2R* (electronic supplementary material, table S1). Intriguingly, most of the recently identified factors required to maintain the oncogenic state in TAM cells [39], change its expression back to NT condition, suggesting a partial reversion of TAM to NT phenotype (figure 3*g*).

To confirm the RNA-Seq results, a set of randomly selected genes was analysed by RT-PCR and exhibited the same behaviour as shown by the RNA-Seq data (electronic supplementary material, figure S5*a*). In addition, we explored whether a similar response to TPL occurs in other breast cancer cell lines. For that, we analysed the expression of these genes in the estrogen receptor-positive MCF-7 line as well as in the triple-negative MDA-MB-231 line and found that 12 of 14 genes had similar behaviour as the MCF10A-ErSrc (electronic supplementary material, figures S5 and S7). These results indicate that in general, several breast cancer cell lines have a similar gene expression response to TPL. Intriguingly, the MCF-7 line is the most sensible to TPL and the MDA-MB-231 the less sensitive (electronic supplementary material, figure S5*b*).

Altogether our data indicate that XPB inhibition by TPL differentially modulates many genes in NT and TAM cells. Unexpectedly, the RNA-Seq results indicate that even when TPL affects RNAPII transcription, some genes are upregulated in response to this insult. This result suggests that there are genes for which, transcription may continue or even increase under conditions in which the levels of RNAPII are reduced as well as the XPB–p52–p8 submodule.

## 2.4. Promoter occupancy and elongating RNAPII are maintained in genes upregulated in response to TPL

The increase in the RNA levels of numerous genes in response to TPL can be a result of different factors, including enhancement of transcription and/or an increase in the accumulation of some RNAs by reducing their degradation. Reports using TPL to analyse pausing at promoters have been described [40,41]. However, the effect of TPL on RNA levels under conditions when the level of RNAPII is reduced was not determined in any of these studies. Therefore, we sought to analyse the genomic occupancy of RNAPII by chromatin immunoprecipitation followed by next-generation sequencing (ChIP-Seq) in NT and TAM cells incubated with or without TPL under similar conditions as those used in the RNA-Seq experiments.

Overall, the metagenome analysis showed that the occupancy of RNAPII on the promoters was higher in TAM cells than in NT cells. As expected, we observed a reduction in the occupancy of RNAPII in NT and TAM cells when treated with TPL, with the main peak that corresponds to paused RNAPII in TPL-treated cells displaced to the 5′ end of the transcription start site (TSS) (figure 4*a*). This data shows that most RNAPII accumulates in the initiation state (PIC) and is in agreement with previous results on the effect of TPL on RNAPII occupancy on gene promoters [40,41]. Also consistent with the existence of highly stable paused RNAPII at some promoters, maintaining the RNAPII in a paused position even after transcription inhibition by TPL [40], since promoters corresponding to paused RNAPII were still detected (figure 4*a*). However, in our experiments, we identified stably paused RNAPII in some promoters in cells with a substantial reduction in the levels of RNAPII and the XPB–p52–p8 submodule of TFIIH. Additionally, some promoters exhibited an increase in RNAPII occupancy in NT (5.42%) and TAM (3.93%) cells incubated with TPL (figure 4*b*; electronic supplementary material, table S2).

Since the RNA-Seq results indicated that some transcripts were upregulated by TPL, we assessed whether there was a correlation between the occupancy level of RNAPII on the promoters with transcripts that were up- or downregulated in response to TPL. As shown in figure 4*c*, in TPL-treated

royalsocietypublishing.org/journal/rsob    Open Biol. **10**: 200050

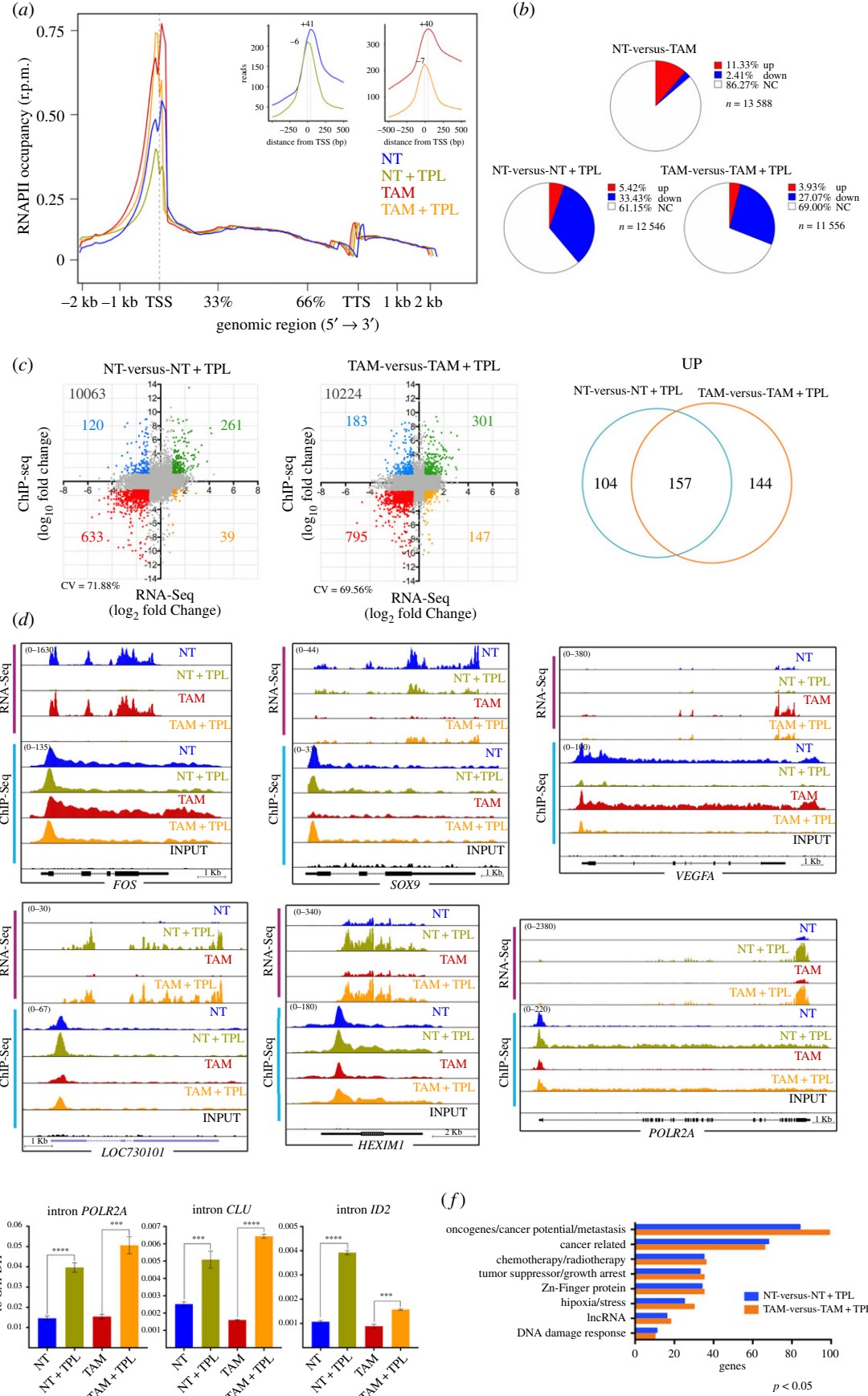

**Figure 4.** (*Caption overleaf.*)

NT and TAM cells, 261 and 301 genes respectively, were found to have an increased association of RNAPII to their promoters, and that correlated with a significant increase in their corresponding RNA levels (electronic supplementary material, table S2). Of these genes, 157 were shared between NT and TAM cells (figure 4c). Also, an increase of RNAPII in the body of these genes is observed (figure 4d).

The correlation analyses indicated that there were different gene expression patterns in response to TPL (figure 4c). There were genes such as *FOS*, on which paused RNAPII was cleared,

**Figure 4.** (*Overleaf.*) Analysis of the positioning of the RNAPII in cells treated with TPL. ChIP-Seq experiments were performed from two independent biological replicas for each condition. (*a*) Meta-analysis of the positioning of the RNAPII comparing the non-treated (NT, blue), non-treated + triptolide (NT + TPL, green), Treated with tamoxifen (TAM, red) and Treated with tamoxifen + triptolide (TAM + TPL, orange) treated with TPL for 4 h at 125 nM. The upper right panel shows the displacement suffered by the cells when are treated with TPL towards the TSS-5'. Note the presence of two RNAPII picks, without TPL the major pick corresponds to paused RNAPII and with TPL the major pick corresponds to initiating RNAPII. Also, note that after TPL treatment in some promoters RNAPII is maintained paused. (*b*) Relation of the occupancy of the RNAPII in gene promoters after the TPL treatment in NT, NT + TPL, TAM and TAM + TPL cells. In red is indicated the promoter occupancy percentage that increases, in blue percentage that goes down and in white promoters that do not have a significant change. (*c*) Correlation between the RNA-Seq data (x-axis) and ChIP-Seq data (y-axis), from the NT-versus-NT + TPL and TAM-versus-TAM + TPL cells. In green are genes that increase both in transcriptional level and RNAPII positioning in the promoter; in blue genes that increase in transcriptional level but have low positioning in the promoter; in red are the double negative genes, genes with lower transcriptional level and lower positioning in the promoter, while in yellow are genes with low transcript level but increase in the positioning of the polymerase. Correlation value (CV). The right panel shows the number of genes that increase in RNA-Seq and ChIP-Seq, unique and common between NT-versus-NT + TPL cells compared to TAM-versus-TAM + TPL. (*d*) Examples of the different behaviours observed in the RNA-Seq (pink bar) and ChIP-Seq (blue bar) FOS, SOX9, VEGFA, LOC730101, HEXIM1 and POLR2A. Images are from the Genome Browser. (*e*) qRT-PCR of intronic sequences to quantify the levels de-novo synthetized mRNAs for genes overexpressed in response to TPL. POLR2A corresponds to the RNAPII large subunit gene, CLU is the clustering gene and ID2 corresponds to the inhibitor of DNA binding gene. In the three genes, an internal sequence of the first intron was evaluated. Data represent three biological and technical replicates. The graph shows mean values ± s.d. (standard deviation). Significant differences were analysed by t-test. Statistical significance is indicated (****$p < 0.0001$ or ***$p < 0.001$). (*f*) Gene ontology of cancer-related genes that were upregulated in the RNA-Seq and the RNAPII occupancy in ChIP-Seq data. Representative plots and graphs in panels (*a*) to (*d*) and (*f*) summarizing the results of two biological replicates from ChIP-Seq analysis.

but initiating RNAPII was enriched and whose transcript levels were reduced by TPL in NT and TAM cells (figure 4*d*; electronic supplementary material, table S2). For other genes, such as *SOX9*, the mRNA levels decreased in the presence of TPL in NT cells, but in TAM cells, TPL induced a significant increase in the RNA levels as well as an increase in RNAPII promoter occupancy of these same genes (figure 4*d*; electronic supplementary material, table S2). The *VEGFA* gene was overexpressed in TAM cells, with high levels of RNAPII in the body of the gene, but TPL repressed its expression, reducing the occupancy of RNAPII (figure 4*d*). The most intriguing response to TPL occurred in genes that were upregulated, such as *LOC730101* and *HEXIM1*, in which RNAPII occupancy on the promoter and along the body of the gene were increased (figure 4*d*). Interestingly, the effect of TPL on transcription induced potent overexpression of the gene that encodes for the large subunit of RNAPII (*POLR2A*) (figure 4*d*). We confirmed that the increase in levels of the mRNAs is due to an enhancement of transcription in response to TPL by evaluating the *novo* transcription of the first intron pre-mRNA of *POLR2A*, *CLU* and *ID2* genes by qRT-PCR (figure 4*e*).

Next, we performed a gene ontology analysis focusing on cancer-related genes in which RNAPII occupancy on the TSS, as well as its RNA levels, was increased (figure 4*f*; electronic supplementary material, table S3). A large number of genes that function as tumour and growth suppressors, such as *PHACTR4* and *ARID4A* [42,43], as well as genes that correspond to chemotherapy/radiotherapy response genes, such as *SNAI1* and *SNX1* [44,45], were overexpressed. However, a large number of genes considered as oncogenes or that promote tumour growth and/or metastasis, such as *SKI*, *EGR4*, *SNAI1* and *CEMIP2* [45–48], were also overexpressed in response to TPL (figure 4*f*; electronic supplementary material, table S2).

To confirm whether genes exhibiting an increase in RNAPII promoter occupancy were also overexpressed in response to TPL and for further analysis, we selected five genes with different cellular functions for qRT-PCR. The selected genes were *ID2*, a transcription factor known to participate in epithelial–mesenchymal transition [49]; *CRY2*, a circadian repressor involved in MYC turnover [50]; long non-coding RNA (lncRNA) *LOC730101*, which is induced

by hypoxia and has been related to metastasis [51]; and *HEXIM1*, an inhibitor of p-TEFb, linked to chemotherapy resistance and that it has been shown that also is overexpressed in response to JQ1 [52,53]. In addition, we analysed *EPAS1*, also known as *HIF-2A*, a transcription factor that responds to hypoxia, a hallmark of cancer. *EPAS1* is overexpressed in TAM cells and its transcript levels were high and maintained in cells incubated with TPL [49]. For all genes, the increase in RNA levels was confirmed (figure 5*a*–*e*). Kinetic analysis of the encoded products of these genes by Western blot confirmed that not only the accumulation of the RNA increases, but also the corresponding protein in response to TPL (figure 5*f*). Intriguingly, at 8 h after TPL treatment, the protein levels decline, due that most of the cells start to die. Furthermore, we verified the increase in the expression of these genes in response to TPL in other breast cancer cell lines by RT-PCR and qRT-PCR. Interestingly, these genes were also overexpressed in the MCF-7 and MDA-MB-231 breast cancer cell lines in response to TPL, indicating that this response is not exclusive to the MCF10A-ErSrc line (electronic supplementary material, figure S7). Collectively, our results indicate that both NT and TAM cells respond to a TPL insult by inhibiting RNA transcription. Yet, despite TPL directly affects transcription initiation, and significantly reduces the levels of RNAPII as well as of the XPB–p52–p8 submodule components of TFIIH, the transcriptional stress imposed to the cells results in activation of selected genes. Among those, genes that suppress cancer growth are overexpressed, but importantly, genes that promote carcinogenesis, chemotherapy resistance and metastasis are also upregulated.

## 2.5. THZ1 drives a similar gene response as TPL in cancer cells

Next, we decided to explore whether incubation with THZ1 induces a similar gene response as TPL. The first approach to answer this question was to determine, by RT-PCR, the response of some of the upregulated genes in response to TPL in NT and TAM cells when incubated with THZ1. For this experiment, we used a concentration of 300 nM THZ1

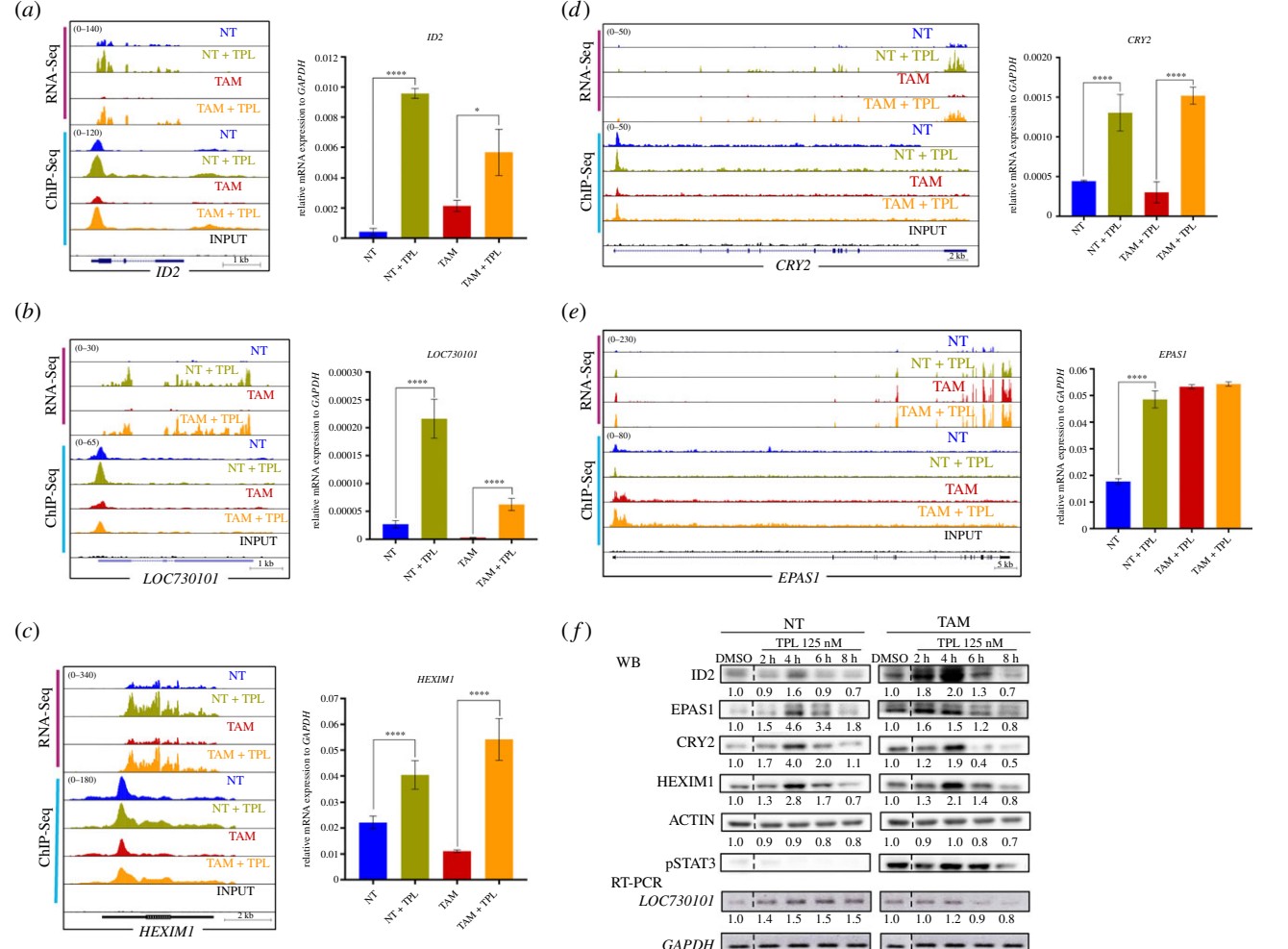

**Figure 5.** Analysis of overexpressed genes and its protein products in response to TPL treatment. (*a*) *ID2* gene, representation of the Genome Browser, RNA-Seq (pink bar) and ChIP-Seq (blue bar), comparing the NT (blue), NT + TPL (green), TAM (red), TAM + TPL (orange) and the INPUT (black). qRT-PCR corroboration is also shown. (*b*) lncRNA *LOC730101* representation in the Genome Browser. qRT-PCR experiments are also show confirming the increase of this transcript after TPL treatment. (*c*) *HEXIM* Genome Browser image. qRT-PCR experiments confirming HEXIM1 overexpression by TPL. (*d*) *CRY2* gene increase with the TPL treatment shown in the Genome Browser image, confirmation by qRT-PCR is also shown. (*e*) *EPAS1* Genome Browser image and qRT-PCR corroboration. (*f*) Example of a time-course experiment of the overexpression of the ID2, HEXIM1, CRY2 and EPAS1 proteins in response to TPL by Western blot analysis. STAT3 phosphorylated is an indicator of the transformed phenotype in TAM cells. A time-course analysis of the overexpression of the *LOC730101* lncRNA by RT-PCR is also shown. The dashed line indicates where the images were cut in order to avoid another treatment time (30 min) included in the experiment. Densitometric analyses were performed using actin or *GAPDH* as a loading control in Western blot and RT-PCR, respectively; the relative quantification is indicated under each blot and the data showed as a representative example from three biological replicates. Data from (*a*) to (*e*) represent three biological replicates. Graphs show mean values ± s.d. (standard deviation). Significant differences were analysed by *t*-test. Statistical significance is indicated (\**p* < 0.05 or \*\*\*\**p* < 0.0001).

for 2 h (figure 6*a*). Figure 6*b* shows that for *ID2, lncRNA LOC730101, HEXIM1* and *EPAS1*, there was a clear increase in the mRNA levels in NT cells incubated with THZ1 (figure 6*b*). In TAM and NT cells, the mRNA levels of *ID2* and the lncRNA increased and similar behaviour of *EPAS1* as that seen with TPL incubation was observed (figure 6*b*). By contrast, no clear increase in the *HEXIM1* transcript level was observed in TAM cells, and in all cell types, *CRY2* level remained the same in THZ1-treated cells. These results suggest that there are some similarities in the gene expression responses to TPL and THZ1 in NT and TAM cells, but that the responses are not identical.

To explore how similar the gene expression responses to TPL and THZ1 are, we used public RNA-Seq data from an oesophageal cancer cell line treated with THZ1 [54]. We found that 94 genes upregulated by TPL that correlated with an increase in the occupancy of RNAPII in the oncogenesis model used here were also upregulated by THZ1 in the

oesophageal cancer cell line (figure 6*c*). These results suggest that in response to transcriptional stress by either TPL or THZ1, a similar set of genes is overexpressed.

## 2.6. Depletion of transcripts encoded by genes overexpressed in response to transcriptional stress augments the cells sensitivity to triptolide

The fact that some genes are overexpressed in response to TPL underscored the possibility that some of them were targets for tumour cell killing. To explore the likelihood that depleting the transcripts of genes overexpressed in response to TPL potentiates the killing effect of TPL, we chose to evaluate the effect of silencing *ID2, CRY2, HEXIM1, LOC730101* and *EPAS1* on cell viability and proliferation. NT and TAM cells were transfected with pooled siRNAs against each of these transcripts for 24, 48 or 72 h, followed by incubation

royalsocietypublishing.org/journal/rsob   Open Biol. **10**: 200050

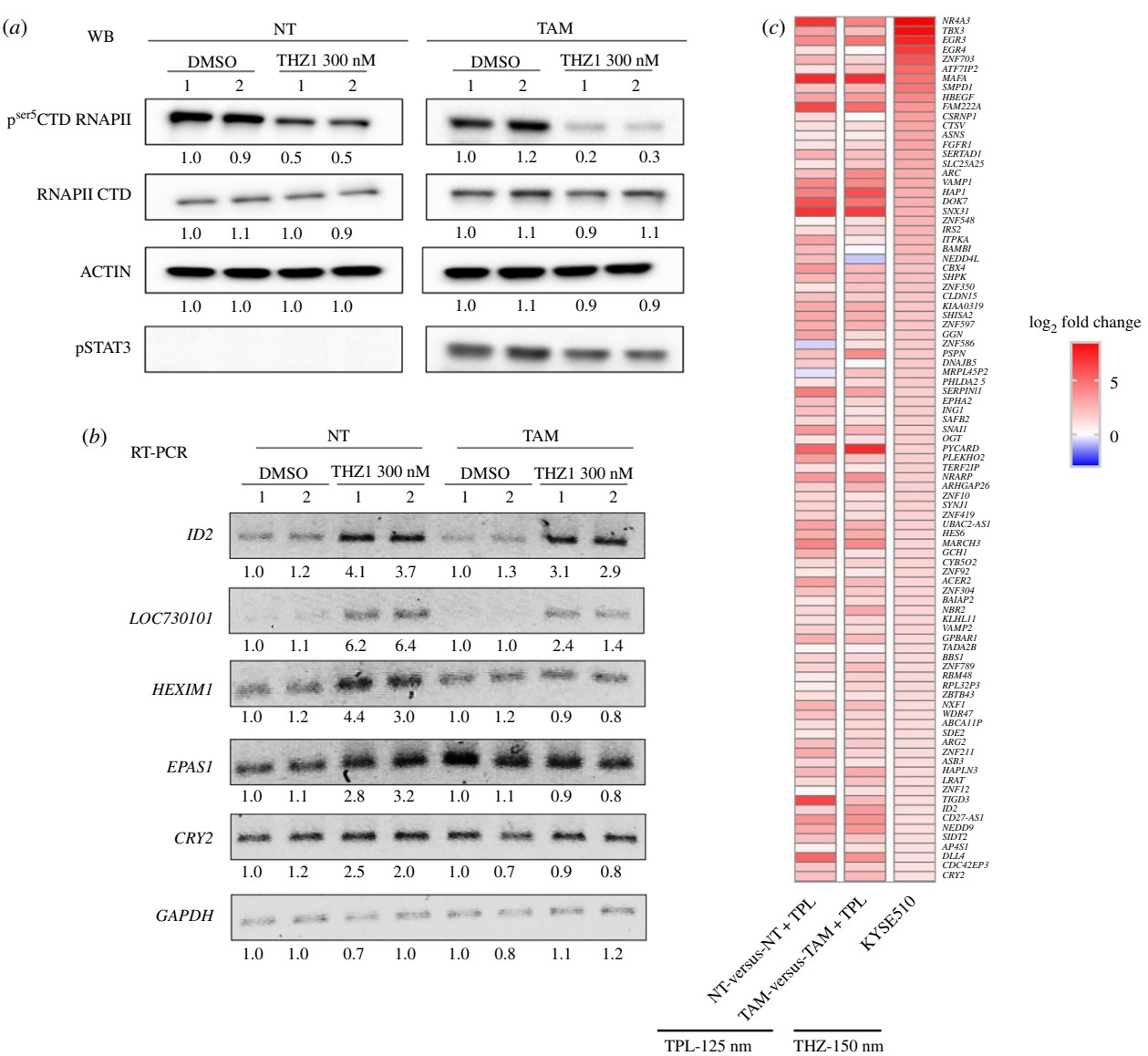

**Figure 6.** THZ1 generates a similar response as observed by the effect of TPL on transformed cells. (a) p$^{Ser5}$CTD RNAPII and RNAPII CTD levels in tamoxifen-treated (TAM) and non-treated (NT) cells treated with 300 nM of TPL for 2 h or with DMSO as control. STAT3 phosphorylated is an indicator of the transformed phenotype in TAM cells. (b) Semiquantitative RT-PCR of NT and TAM cells treated with 300 nM of THZ1 for 2 h of ID2, LOC730101, CRY2, EPAS1 and HEXIM1. Densitometric analyses (a) and (b) were performed using actin or GAPDH as a loading control in western blot and RT-PCR respectively; the relative quantification is indicated under each blot and the data showed represent at less two biological replicates. (c) Comparison with published data of the THZ1 response in KYSE510 cells [54] indicating a similar response to some of the overexpressed genes after TPL treatment.

with 125 nM TPL for 4 h. Consistent with data shown above, 125 nM TPL alone or in combination with the scrambled siRNAs did not affect NT and TAM cells' survival (figure 7, middle panels). All targeted RNAs were effectively silenced by the corresponding siRNAs, with the maximum reduction at 72 h post-transfection, after that, the cells were incubated with 125 nM of TPL for 4 h and evaluated by immunoblot and flow cytometry (figure 7a–e, left panels). Depletion of the mRNAs encoding for ID2, CRY2 and HEXIM1 significantly reduced the viability of TPL-treated NT and TAM cells as compared to cells incubated with scrambled siRNAs or TPL alone (figure 7a–c, middle panels). Different from ID2, CRY2, HEXIM1 depletion, silencing lncRNA LOC730101 resulted in diminished cell viability, even in the absence of TPL, but the effect of silencing this lncRNA, intensified the killing capacity of TPL in both NT and TAM cells (figure 7e, middle panel). Surprisingly, reducing the levels of EPAS1 (HIF-2A) RNA did not result in the enhanced killing capacity of TPL (figure 7d, middle panel). Unlike NT

cells, the majority of surviving TAM cells underwent at least one round of proliferation less than NT transformed cells following RNA silencing of all genes (including EPAS1) and TPL treatment, or even two (figure 7a–e, right panels), In summary, these results show that depletion of some genes overexpressed in response to TPL sensitizes cells to TPL treatment; therefore, these genes are possible targets to enhance the killing effect of TPL on cancer cells.

# 3. Discussion

The use of different chemical and physical agents is still the most common approach for killing cancer cells. The mechanism of action of chemotherapy drugs will directly kill cancer cells or stop their proliferation by inducing a cellular response linked to the mechanism of action of the drug. Substances such as TPL and THZ1 that directly impact the activities of TFIIH have a high potential for use in cancer treatment.

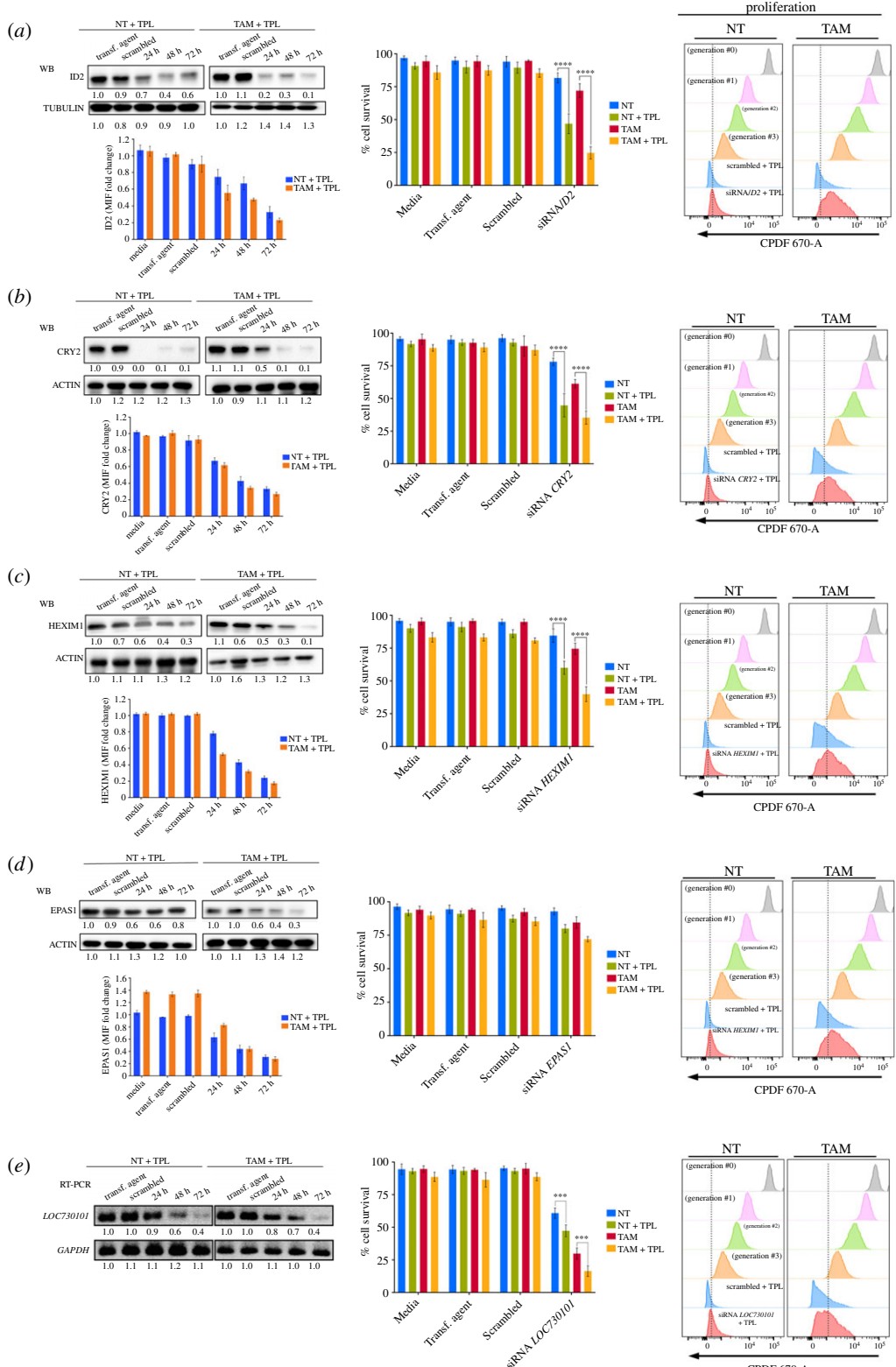

**Figure 7.** Depletion of transcripts of overexpressed genes by transcriptional stress enhances the sensitivity to TPL. *ID2, CRY2, HEXIM1, EPAS1* and *LOC730101* were depleted by incubating the corresponding siRNAs at different times. After that, the cells were incubated with 125 nM of TPL for 4 h. The depletion of each protein product was determined by Western blot analysis and by cytometry. In the case of the *LOC73101* transcript, its depletion was analysed by RT-PCR. The effect on cell survival and proliferation were also analysed by flow cytometry and the results indicated in the figure. (*a*) ID2 analysis. (*b*) CRY2 analysis. (*c*) HEXIM1 analysis. (*d*) EPAS analysis. (*e*) LOC730101 analysis. Cell survival and proliferation only show the effect of TPL after incubation of the corresponding siRNA by 72 h. The name of each gene is indicated in the corresponding panel. Densitometric analyses were performed using actin or *GAPDH* as a loading control in western blot and RT-PCR, respectively; the relative quantification is indicated under each blot and the data showed as a representative example from three biological replicates. Graphs show mean values ± s.d. (standard deviation). Significant differences were analysed by *t*-test. Statistical significance is indicated (****$p < 0.0001$ or ***$p < 0.001$).

In this study, we performed a systematic analysis of the effects and response to these drugs, primarily TPL, in the MCF10A-ErSrc oncogenesis model. Our results strongly suggest that TAM tumour cells exhibited increased sensitivity to TPL or THZ1 as compared to their NT counterparts and that the combination of both drugs had a synergic effect on cell death. From a mechanistic point of view, we found that even though TPL and THZ1 directly affect transcription

initiation by RNAPII of the majority of genes, specific genes are overexpressed as a result of the transcriptional stress to which the cells are submitted upon treatment with these drugs, underscoring the possibility to target these genes in conjunction with TFIIH inhibitors to kill cancer cells.

In this oncogenic model, both TPL and THZ1 were found to kill cells via apoptosis, stop proliferation one replication cycle earlier in TAM cells than in NT cells and arrest cells in $G_1$. Co-treatment with TPL and THZ1 had a synergic effect on cell viability. Interestingly, the combination of TPL and THZ1 killed preferentially the transformed (cancerous) cells with high efficacy. These results also suggest that the simultaneous use of substances that affect different TFIIH functions is an interesting alternative to treat cancer and opens the possibility of searching for new drugs that may affect other TFIIH subunits, particularly if we consider the hepatotoxic effect of TPL and its derivatives [55].

Inhibition of transcription by TPL induces the proteasome-dependent degradation of RNAPII [56,57]. It has been proposed that TPL induces the degradation of the polymerase following phosphorylation of POLR2A, the largest subunit of RNAPII by CDK7 [56]. We report here that incubation with TPL induced the degradation not only of RNAPII but also of XPB as well as of p52 and p8, but it did not affect other TFIIH subunits. THZ1, which inhibits the kinase activity of CDK7, did not have any effect on the RNAPII and TFIIH protein levels.

It is known that the interaction of XPB with p52 and p8 modulates its ATPase activity [31,33,58], and the recently solved structure of the TFIIH core shows that the N-terminal domain of XPB and the clutch domain of p52 have a similar fold and form a symmetric dimer. This interaction is not close to the ATPase domain, suggesting that regulation of ATPase activity occurs through the combined interaction of p52 and p8 with XPB [34]. Accordingly, our findings show that the covalent binding of TPL to the ATPase domain of XPB destabilizes its interaction with p52, hindering the assembly of the subunits. Furthermore, our computational study allowed us to propose that the dissociation/degradation of the XPB–p52–p8 submodule is mainly caused by the separation of XPB HD1–HD2 induced by the presence of TPL at the domain interface. Additionally, these results suggest that in the context of TFIIH, XPB, p52 and p8 form a submodule that is stable only when the three subunits are interacting and that the other TFIIH modules are not affected in its absence. This idea is consistent with the conformation and organization model described in the recently solved structure of the TFIIH core [34].

Intriguingly, degradation of RNAPII, as well as the XPB–p52–p8 submodule was accelerated in TAM cells incubated with TPL. Thus, it is feasible the mechanism that increases the sensitivity of TAM cells to TPL is partially due to the high proliferation rate of these cells, the turnover of these proteins is not fast enough to compensate for RNAPII depletion and, as a consequence, the effect on global transcription is enhanced.

Several reports on the effect of TPL using PRO-Seq and GRO-Seq have shown that the immediate effect is the reduction in transcription initiation of at least 90% of genes [22,23]. These studies were performed after short incubation times, with extremely high concentrations (10–500 µM) of TPL, but when the levels of RNAPII were not affected. We analysed the transcriptome of NT and TAM cells under conditions in which the levels of RNAPII were reduced by half,

with a minor effect in cell viability. However, we detected that many transcripts were downregulated and many others were upregulated. These results correlate with our data in *Drosophila*, in which both up- and downregulated transcripts were observed in p8 and p52 mutant organisms [37,38]. Interestingly, the increase in the levels of several specific genes in response to TPL also occurred in other breast cancer cell lines. In similarity with our results, it has been documented that UV irradiation causes the degradation of the RNAPII, but with the remaining RNAPII, the cell overexpresses specific genes a response to this insult. A similar situation may be operating as a response to TPL [59]. Furthermore, our ChIP-Seq results showed that the increase in the RNA levels of many genes was correlated with an increase in the occupancy of RNAPII on the corresponding promoters as well as in the gene bodies and qRT-PCR of intronic sequences confirmed that it is due to an enhancement of transcription. This is supported by the fact that the corresponding protein products of the genes analysed here also increase after the TPL treatment. An explanation for this phenomenon could be that, as recently reported [60], TPL inhibits transcription by its interaction with XPB, but if XPB is not present, then transcription is not inhibited. In support of this hypothesis, we found that TPL induced degradation of the XPB–p52–p8 submodule. Although it is feasible that under our experimental conditions, some genes were not affected by TPL, probably because some genes do not require XPB as it has been shown in yeast [15], this hypothesis is not supported by the observation that the response is the overexpression of specific genes, with a significant increase in the corresponding RNA levels and protein products. In addition, we observed that numerous genes upregulated by TPL were also upregulated in cells treated with THZ1, which inhibits CDK7 but does not affect the levels of the TFIIH subunits or RNAPII. Therefore, our data evidence that TPL and THZ1 activate a mechanism of gene response to transcriptional stress in treated cells.

TPL is a very effective substance for killing cancer cells, and related compounds are now in clinical phases of development. However, one of the problems with the use of TPL in patients is that it is highly toxic, and off-target effects cannot be ignored [20,55]. Therefore, finding new targets to enhance the effect of reduced concentrations of TPL is very attractive. Here, we analysed only five of many other possible targets found to be overexpressed in response to TPL and in four, its depletion enhances the toxic effect of TPL at low concentrations. These results suggest that many of the other gene products overexpressed in response to TPL may improve the anti-tumoural capacity of TPL, opening a new avenue to complement the attack on the transcriptional addiction of cancer cells.

Although treatment with TPL at high doses eventually killed most TAM cells, the circumstances may be different under *in vivo* conditions, and it is possible that many of the cancerous cells in the tumour are exposed to lower concentrations of TPL, allowing them to respond to transcriptional stress, by upregulating the transcription level of some of the genes we report. This gene response is relevant for the treatment of tumours by TPL or THZ1, more so considering that some genes that we found to be over-activated encode factors that promote tumourigenesis and/or metastasis and suggest that transformed cells might rapidly develop resistance to TPL/THZ inhibitors.

In conclusion, the results presented here show that cells have the capacity to respond to the transcriptional insult

royalsocietypublishing.org/journal/rsob   Open Biol. **10**: 200050

caused by TPL by overexpressing some genes. Some of these genes are also overexpressed in response to THZ1, and these genes are possible targets in combination with TPL to preferentially kill cancer cells. However, this study also invokes several questions. For instance, does the depletion of genes overexpressed in response to transcriptional stress also enhance the effect of THZ1 on cancer cells? What is the mechanism that potentiates the effect of TPL by depleting genes that have different functions, such as *ID2*, *HEXIM1* and *CRY2*? Is the overexpression of some genes by TPL is achieved via only one specific response pathway, or are many pathways involved? The answers to these questions will be relevant in understanding the response to chemotherapy based on transcription inhibitors used in cancer.

# 4. Material and methods

## 4.1. Cell lines

MCF10A-ERSrc cell line was donated by Dr Struhl. Cells were treated with 2 µM tamoxifen [28] and morphological transformation is observed within 36–72 h. MCF10A and MCF10A-ER-Src cells were cultured in DMEM/F12 medium supplemented with: 5% donor horse serum (charcoal stripping fetal bovine serum was prepared following Struhl's group protocol for the MCF10AEr-Src cell line (https://www.encodeproject.org/documents/ae279a0c-4d69–4d46-afa1-33d9657c0ea2/@@download/attachment/MCF-10A_Struhl_protocol.pdf), while the heat inactivated serum was employed for the MCF10A line), 20 ng ml$^{-1}$ epidermal growth factor, 10 µg ml$^{-1}$ insulin, 100 µg ml$^{-1}$ hydrocortisone, 1 ng ml$^{-1}$ cholera toxin, 1% pen/step, with the addition of puromycin (only in the line MCF10AEr-Src) [28,61]. MDA-MB-231, MCF7 and HEK-293 cells lines were grown in DMEM, 10% FBS and 1% pen/step [36,62].

## 4.2. Chemicals

Tamoxifen, 4-Hidroxytamoxifen (Sigma, Cat. H790); Triptolide (Tocris Cat. 3253) and THZ1 (APExBIO, Cat. A8882).

## 4.3. Western blotting

Cell extracts were prepared as described in Gurrion *et al.* [63]. Antibodies used were: 8WG16, H14, Phospho-STAT3, TBP, CDK7, MAT1, CYCLIN H, XPB, p52, p62, XPD, p8, EPAS, ID2, CRY2, HEXIM1, ACTIN and TUBULIN (electronic supplementary material, table S4). Horseradish peroxidase-coupled secondary antibodies (Invitrogen, 1 : 3000) were used for chemiluminescence detection through Thermo Scientific Pierce ECL. The images were taken with the Gel Doc XR+ system (Bio-Rad). Experiments were performed by at less three independent biological replicates.

## 4.4. RT-PCR or qRT-PCR

Total RNA was extracted with TRizol (Invitrogen) and equal quantity from each sample was used. cDNA synthesis was performed in a reaction mix containing 1 µg of total RNA, oligo-dT, random primer and M-MLV Reverse Transcriptase (Invitrogen). For the intron cDNA same condition were used, except for 3 µg of total RNA and specific oligo. qPCR analyses were performed with LightCycler FastStart DNA Master$^{PLUS}$ SYBR

Green I and the LightCycler 1.5 Instrument (Roche). Relative expression level of each analysed gene was calculated by $2^{-\Delta Ct}$, where $\Delta Ct = (Ct \text{ target gene} - Ct \text{ control gene})$, using *GAPDH* as an internal control [64]. Transcript abundance quantification and DNA number copies were measured by triplicate and three independent biological replicates were analysed. Primers are described in electronic supplementary material, table S4.

## 4.5. siRNA assays

siRNA-silencing was performed according to Dharmacon instructions. The siRNAs used were: *CRY2* (Cat. L-014151-01-0010), *EPAS1* (Cat. L-004814-00-0010), *HEXIM1* (Cat. L-012225-01-0010), *ID2* (Cat. L-009864-00-0010), *LOC730101* (Cat. R-189565-00-0010) and Scrambled (Cat. D-001810-10-20). siRNAs were used in a 25 and 50 nM final concentration for 24–72 h and then mRNA or protein analysis were performed by western blot and flow cytometry. Experiments were performed by at less three independent biological replicates.

## 4.6. Flow cytometry

For proliferation assays, cells were loaded with the Cell Proliferation Dye eFluor 670 (Invitrogen, Cat. 65-0840). Viability and Apoptosis assays were according to manufacture: Fixable Viability Dye eFluor 780 (Invitrogen, Cat. 65-0865) and Biolegend Pacific Blue Annexin V or FITC Annexin V (BioLegend, Cat. 640918 or Cat. 640906, respectively). Intracellular protein staining was performed as previously described [65]. Cell cycle assays were performed staining the cells with DAPI 5 µg ml$^{-1}$ in PBS for 30 min at 37°C. Samples were acquired on a BD FACSCanto II or BD FACs Aria Fusion flow cytometer with the BD FACSDiva software and analysed using the FlowJo v10.5.3. Experiments were performed by three independent biological replicates.

Cell viability data was analysed for cooperative effects between TPL and THZ1 using the method implemented by Chou & Talalay [30,66]. CI and Fa values were calculated using CompuSyn software (available for free download from www.combosyn.com). CI values = 1 indicate an additive effect, and C < 1 and C1 > 1 indicate synergism and antagonism, respectively.

## 4.7. Split-GFP assays

Plasmids pCNV_GFP1-9-OPT, pcDNA_GFP10-GCN4 and pcDNA_GCN4-GFP11, were donated by Dra. Cabantous [36]. Stable transfection was performed in the HEK-293 cells of pCNV_GFP1-9-OPT plasmid using lipofectamine 3000. GCN4 sequences were removed with the restriction enzymes *Bspe*I:*Xba*I for pcDNA_GFP10-GCN4 and *Not*I:*Cla*I for pcDNA_GFP11-GCN4. *p52* (NM_001517) and *XPB* (NM_000122) were amplified from cDNA and inserted into *Mre*I:*Xba*I and *Not*I:*Cla*I cloning sites of pcDNA_GFP10 and pcDNA_GFP11 vectors, respectively.

HEK-293_GFP1–9 cells were co-transfected with 1.5 mg of each plasmid: p52–GFP10 and GFP11–XPB. Thirty-six hours after transfection, cells were visualized in the Olympus FV1000 Multi-photonic confocal microscope 60X. On the other hand, TPL at different concentration was added to co-transfected cells 8 h later and 36 h afterwards cells were stained for viability. As negative control the plasmids GCN4-GFP10

and GFP11-GCN4 with the same conditions were used. Experiments were performed by at less three independent biological replicates.

## 4.8. RNA-Seq and bioinformatic analysis

Here $8 \times 10^6$ cells were treated with TPL 125 nM or DMSO 4 h. Experiments were performed by two independent biological replicates. Total RNA was extracted with TRIzol (Invitrogen) according to manufacturer's protocol. Poly-A enriched RNA was sequenced using Illumina HiSeq™ 2000 by the Beijing Genomics Institute (BGI). Briefly, mRNA enrichment was performed using oligo(dT), mRNA was fragmented and were used as template for the synthesis of cDNA by reverse transcription. The Agilent 2100 Bioanaylzer and ABI StepOne-Plus Real-Time PCR System were used in quantification and qualification of the sample libraries. Libraries were sequenced in Illumina HiSeq2000. Bowtie2 v. 0.9.6 [67], was used to map clean reads to reference gene and BWA v. 0.7.13 [68] to reference genome *hg19*. Sequencing reads were checked with FastQC v. 0.11.7. Expression levels were quantified using FPKM by RSEM v. 1.3.0 [69]. Differential expression data were filtered using FDR $\leq 0.001$ [70], and $\log_2$ FC $\geq 1.2$. Gene expression analyses were performed using scripts in R v. 3.5.1. Analysis details are available upon request.

## 4.9. ChIP-Seq and bioinformatic analysis

Cells were treated with TPL 125 nM or DMSO for 4 h. ChIP-Seq was performed according to a previously published protocol [71] and experiments were performed by two independent biological replicate (electronic supplementary material, figure S6). Briefly, cells were cross-linked with 1% PFA at room temperature for 10 min and sonicated for 11 cycles (30 s ON/OFF, Diagenode Bioruptor Pico bath sonicator). 5% of the lysate was reserved as Input. Lysate was incubated with 5 mg of the antibody (8WG16) or irrelevant rabbit IgG (Invitrogen). Library construction and Illumina sequencing were performed using Illumina HiSeq SE50 platform at BGI.

Briefly, data filtering included removing adaptor sequences, contamination and low-quality reads from raw reads by BGI programs, like SOAPnuke filter. Clean data were mapped to the reference genome (*hg19*) by SOAPaligner/SOAP2 [72]. MACS2 v. 1.4.2 [73] was used to call peaks and generate Bedgraph files that show fold change enrichment over input. Bedgraph files were then converted into BigWig files and uploaded to UCSC Genome Browser for visualization.

HOMER (Next-Generation Sequencing Analysis; *annotatePeaks.pl*) [74] was used to annotate the peaks to the genome.

Difference in RNAPII enrichment between TPL-treated and control cells we performed a differential binding analysis using HOMER (getDiffExpression.pl). Differential expression data were filtered using $\log_2$ FC $\geq 1$. Graphic representation of the data was performed using R v. 3.5.1 and GraphPad Prism v. 7. Analysis details are available upon request.

## 4.10. Correlation of ChIP-Seq and RNA-Seq analysis

To perform a correlation between RNA-Seq and ChIP-Seq analyses, we considerate only the RNAPII peaks located a $\pm 1$ kb window spanning the Transcription Start Site (TSS). Differential expression data were filtered using $\log_2$ FC $\geq 1$ for ChIP-Seq and $\log_2$ FC $\geq 1.2$ for RNA-Seq. Graph was made GraphPad Prism (figure 4*c*). Correlation value (CV) represents the similarity degree between ChIP-Seq and RNA-Seq data.

Accession number for raw and processed ChIP-Seq and RNA-Seq data reported in this paper is GEO: GSE135256.

## 4.11. Molecular dynamics

The cryo-electron microscopy structure of *Homo sapiens* TFIIH (PDB ID: 6NMI [34]) was retrieved from the Protein Data Bank (figure 2*b*). The isolated XPB–p52–p8 submodule was employed to perform the covalent docking of the optimized TPL structure into Cys342 residue of the XPB component employing the AutoDock v4.2 [75]. The Apo and two holo (TPL-bound) forms of XPB–p52–p8 submodule were submitted to 100 ns all-atom molecular dynamics (MD) simulations using the CHARMM36 force field [76] implemented in the GROMACS 5.1.4 package [77]. (For more details, see electronic supplementary material, *methods* and electronic supplementary material table S5.)

Data accessibility. Accession number for raw and processed ChIP-seq and RNA-seq data reported in this paper is GEO: GSE135256.
Competing interests. We declare we have no competing interests.
Funding. This work was supported by the CONACyT grant no. 1977-Problemas Nacionales program, PAPPIT/UNAM grant no. IN200315 and CONACyT no. 25088 to MZ. M.U.-A. received a scholarship from CONACyT (414127), as a student of the Programa de Doctorado en Ciencias Bioquímicas at the Universidad Nacional Autónoma de México.
Acknowledgements. We thank Arturo Pimentel, Andres Saralegui, Chris Wood and the National Laboratory of Advance Microscopy at the Instituo de Biotecnología/UNAM for advice in the use of the confocal microscopes. L.D. thanks the Dirección General de Cómputo y de Tecnologías de Información y Comunicación for the support received in the use of the HP Cluster Platform 3000SL supercomputer (LANCAD-UNAM-DGTIC-306).

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
