## [Reviewer comments · Open Biology]

Review History

RSOB-20-0050.R0 (Original submission)

Review form: Reviewer 1

Recommendation

Major revision is needed (please make suggestions in comments)

Do you have any ethical concerns with this paper?

No

Comments to the Author

Comments

Uriostegui-Arcos M, et al analyzed the cytotoxic effects of TPL and THZ1 in a MCF10A-ErSrc cell model, revealing that tumor cells are more prone to be targeted by the small molecular inhibitors to TFIIH complex. The authors also proved that TPL specifically interferes with the XPB and p52 in TFIIH complex, resulting in the reduction of many transcripts by RNAseq and RNAPII-Chip-seq assays. Surprisingly, the authors also reported the RNAPII promoter occupancy of a significant number of genes involved in tumor growth and metastasis did not reduce upon TPL

treatment. Such induction of certain genes by TPL can be validated in THZ1 treatment. The depletion of some of these genes can enhance the cytotoxic effects of TPL. The authors made great efforts to dissect the mechanism underlying the cytotoxic effects of TPL and found an interesting potential drug resistant genes. The data are novel and interesting. Below are some concerns

- Majors
- 1) Fig 1A, the author can use Chou-Talalay method for drug combination, which is the best way to demonstrate whether two drugs possess additive or synergistic effects against certain cancer cells.
 - 2) The authors found that the induction of several genes involved in tumor growth and metastasis increases upon the treatment of (125 nM) TPL. When looking into closely, such induced genes are early-response genes to TPL treatment, and the induction gradually fade away as shown in Fig 5f, with some genes even lower than DMSO treatment group. It may be due to the cell death which can be inferred from Fig 1. Hence, it is better to check whether such induction is dose-dependent and low dose of TPL can increase such gene expression within 24 h treatment when the majority of cells are still viable.
 - 3) To exclude potential off-target effects of chemicals, it is needed to check whether such genes also change in the XPB knockdown Tam treated or non-treated MCF10A-ErSrc cells.

Minors:

- 1) Although the authors made a clear explanation for "T cells", an abbreviation for "tamoxifen Treated cells". There is some confusion because the word "T cells" has its own special meaning, commonly referred to an immune cell population. Therefore, I recommend the authors to change another word for "tamoxifen treated cells".
- 2) To change P52 and P8 to p52 and p8.
- 3) Fig 2C, no statistical data on the effects of TPL.
- 4) In the M&M section, please describe the serum used to culture MCF10A-ERSRC cells.
- 5) Fig S1, the analysis of the Annexin V/PI staining is not precise, since the 2nd quadrant for DMSO treatment group is not correct. The author need to use no staining, PI/Annexin VI single staining to divide four quadrants.
- 6) Fig 7, The author described that TPL treatment for 4 h and cells were collected to check knockdown effect. As for the cell survival assay, please indicate the duration for the TPL treatment.

Review form: Reviewer 2

Recommendation

Accept with minor revision (please list in comments)

Are each of the following suitable for general readers?

- a) **Title**
Yes
- b) **Summary**
Yes
- c) **Introduction**
Yes

Is the supplementary material necessary; and if so is it adequate and clear?

Yes

Do you have any ethical concerns with this paper?

No

Comments to the Author

The manuscript by Uriostegui-Arcos and colleagues (RSOB-20-0050) reports a detailed analysis of the gene expression responses of an oncogenesis cellular model to TPL (Triptolide) and THZ1 (phenylaminopyrimidine) two well characterized inhibitors of the transcription/DNA repair factor TFIID.

As expected tumour cells exhibited increased sensitivity to TPL or THZ1 but more importantly, transcriptome analysis combined with chromatin immunoprecipitation demonstrated that although the levels of many transcripts are reduced in TPL-treated cells, a significant number of transcripts increased following TPL treatment, with maintained or increased RNA Pol II-promoter occupancy. These results suggest that cells generate a gene expression response to the stress caused by the inhibition of transcription and reveal possible new targets against cancer. Interestingly, depletion of transcripts encoded by some genes overexpressed in response to transcriptional stress augments the cells sensitivity to TPL.

Overall, the manuscript provides interesting new insights into the regulation of TFIID activities and new perspectives on the use of substances that target the basal transcription machinery. I have however a few concerns that could/should be taken into consideration

Point 1. The authors show that TPL interferes with the XPB-p52 interaction, inducing the degradation of the XPB-p52-p8 submodule of TFIID. The authors should show that as for CAK subunits and XPD, other subunits of core-TFIID are not affected. Also as these experiments were performed in triplicate, the repeated experiments could be quantified to assess the statistical significance of the claim.

Point 2. A confocal analysis of the Split-GFP complementation assays between p52 and XPB (Figure 2c) shows that fluorescence is un-homogenously distributed in the nucleoplasm which is striking as both XPB and p52 are in general evenly distributed. This point should be discussed as the accumulation of the XPB-p52-GFP complex could result from aggregation of non-functional proteins. Is the localization of the GFP fragments GFP10 and GFP11 compatible with the formation of a functional TFIID complex - The distance between the extremities of p52 and XPB fused to GFP10 and GFP11 should be reasonable - ?

Point 3. The authors present molecular dynamic simulation indicating that the presence of TPL in the active site of XPB alters the HD1-HD2 interface which suggests that TPL induces a conformational change leading to the destabilization of the XPB/p52 interface. Although this presentation in the result section is in my opinion valid, the authors should take care to avoid over-interpretations. I would not claim, as written in the discussion, that the MD simulation "allowed us to propose that the dissociation/degradation of the XPB-P52-P8 submodule is mainly caused by the separation of XPB HD1-HD2 induced by the presence of TPL at the domain interface." For such a claim the authors should for example analyze the MD trajectories of residues at the XPB/p52 interface and show the interface is affected by the presence of TPL.

Minor

Point 4. Some labels from Figure 3 are seem confusing and/or are not clearly documented in the Figure legend. In panel (a), was RNA Pol II (as mentioned in the text) or phosphorylated RNA Pol II detected? What does TF in panels (c) and (e) mean?

Point 5. The authors discuss stably paused promoters but do provide a clear definition. Is that proposed by Chen and colleagues (Chen, Genes and Dev, 2014)?

Point 6. The discussion of RNA-Seq and Chip-Seq data for individual genes (mainly in Figure 4d) qualitative and not easy to follow. The authors could annotate the panels from Figure 4d and, for example could indicate the amounts of paused and initiating polymerase.

Decision letter (RSOB-20-0050.R0)

14-Apr-2020

Dear Dr Zurita,

We are writing to inform you that the Editor has reached a decision on your manuscript RSOB-20-0050 entitled "Disruption of TFIID activities generates a stress gene expression response and reveals possible new targets against cancer", submitted to Open Biology.

As you will see from the reviewers' comments below, there are a number of criticisms that prevent us from accepting your manuscript at this stage. The reviewers suggest, however, that a revised version could be acceptable, if you are able to address their concerns. If you think that you can deal satisfactorily with the reviewer's suggestions, we would be pleased to consider a revised manuscript.

Please address the reviewers comments, please focus on addressing the comments that can be addressed without further experiments. You do not need to do any further experiments.

The revision will be re-reviewed, where possible, by the original referees. As such, please submit the revised version of your manuscript within four weeks. If you do not think you will be able to meet this date please let us know immediately.

When submitting your revised manuscript, please respond to the comments made by the referee(s) and upload a file "Response to Referees" in "Section 6 - File Upload". You can use this to document any changes you make to the original manuscript. In order to expedite the processing of the revised manuscript, please be as specific as possible in your response to the referee(s).

Please see our detailed instructions for revision requirements
<https://royalsociety.org/journals/authors/author-guidelines/>

Sincerely,
The Open Biology Team
<mailto:openbiology@royalsociety.org>

Reviewer(s)' Comments to Author(s):

Referee: 1

Comments to the Author(s)

Comments

Uriostegui-Arcos M, et al analyzed the cytotoxic effects of TPL and THZ1 in a MCF10A-ErSrc cell model, revealing that tumor cells are more prone to be targeted by the small molecular inhibitors to TFIIH complex. The authors also proved that TPL specifically interferes with the XPB and p52 in TFIIH complex, resulting in the reduction of many transcripts by RNAseq and RNAPII-Chip-seq assays. Surprisingly, the authors also reported the RNAPII promoter occupancy of a significant number of genes involved in tumor growth and metastasis did not reduce upon TPL treatment. Such induction of certain genes by TPL can be validated in THZ1 treatment. The depletion of some of these genes can enhance the cytotoxic effects of TPL. The authors made great efforts to dissect the mechanism underlying the cytotoxic effects of TPL and found an interesting potential drug resistant genes. The data are novel and interesting. Below are some concerns
Majors

- 1) Fig 1A, the author can use Chou-Talalay method for drug combination, which is the best way to demonstrate whether two drugs possess additive or synergistic effects against certain cancer cells.
- 2) The authors found that the induction of several genes involved in tumor growth and metastasis increases upon the treatment of (125 nM) TPL. When looking into closely, such induced genes are early-response genes to TPL treatment, and the induction gradually fade away as shown in Fig 5f, with some genes even lower than DMSO treatment group. It may be due to the cell death which can be inferred from Fig 1. Hence, it is better to check whether such induction is dose-dependent and low dose of TPL can increase such gene expression within 24 h treatment when the majority of cells are still viable.
- 3) To exclude potential off-target effects of chemicals, it is needed to check whether such genes also change in the XPB knockdown Tam treated or non-treated MCF10A-ErSrc cells.

Minors:

- 1) Although the authors made a clear explanation for "T cells", an abbreviation for "tamoxifen Treated cells". There is some confusion because the word "T cells" has its own special meaning, commonly referred to an immune cell population. Therefore, I recommend the authors to change another word for "tamoxifen treated cells".
- 2) To change P52 and P8 to p52 and p8.
- 3) Fig 2C, no statistical data on the effects of TPL.
- 4) In the M&M section, please describe the serum used to culture MCF10A-ERSRC cells.
- 5) Fig S1, the analysis of the Annexin V/PI staining is not precise, since the 2nd quadrant for DMSO treatment group is not correct. The author need to use no staining, PI/Annexin VI single staining to divide four quadrants.
- 6) Fig 7, The author described that TPL treatment for 4 h and cells were collected to check knockdown effect. As for the cell survival assay, please indicate the duration for the TPL treatment.

Referee: 2

Comments to the Author(s)

The manuscript by Uriostegui-Arcos and colleagues (RSOB-20-0050) reports a detailed analysis of the gene expression responses of an oncogenesis cellular model to TPL (Triptolide) and THZ1 (phenylaminopyrimidine) two well characterized inhibitors of the transcription/DNA repair factor TFIIH.

As expected tumour cells exhibited increased sensitivity to TPL or THZ1 but more importantly, transcriptome analysis combined with chromatin immunoprecipitation demonstrated that

although the levels of many transcripts are reduced in TPL-treated cells, a significant number of transcripts increased following TPL treatment, with maintained or increased RNA Pol II-promoter occupancy. These results suggest that cells generate a gene expression response to the stress caused by the inhibition of transcription and reveal possible new targets against cancer. Interestingly, depletion of transcripts encoded by some genes overexpressed in response to transcriptional stress augments the cells sensitivity to TPL.

Overall, the manuscript provides interesting new insights into the regulation of TFIIH activities and new perspectives on the use of substances that target the basal transcription machinery. I have however a few concerns that could/should be taken into consideration

Point 1. The authors show that TPL interferes with the XPB-p52 interaction, inducing the degradation of the XPB-p52-p8 submodule of TFIIH. The authors should show that as for CAK subunits and XPD, other subunits of core-TFIIH are not affected. Also as these experiments were performed in triplicate, the repeated experiments could be quantified to assess the statistical significance of the claim.

Point 2. A confocal analysis of the Split-GFP complementation assays between p52 and XPB (Figure 2c) shows that fluorescence is un-homogenously distributed in the nucleoplasm which is striking as both XPB and p52 are in general evenly distributed. This point should be discussed as the accumulation of the XPB-p52-GFP complex could result from aggregation of non-functional proteins. Is the localization of the GFP fragments GFP10 and GFP11 compatible with the formation of a functional TFIIH complex - The distance between the extremities of p52 and XPB fused to GFP10 and GFP11 should be reasonable - ?

Point 3. The authors present molecular dynamic simulation indicating that the presence of TPL in the active site of XPB alters the HD1-HD2 interface which suggests that TPL induces a conformational change leading to the destabilization of the XPB/p52 interface. Although this presentation in the result section is in my opinion valid, the authors should take care to avoid over-interpretations. I would not claim, as written in the discussion, that the MD simulation "allowed us to propose that the dissociation/ degradation of the XPB-P52-P8 submodule is mainly caused by the separation of XPB HD1-HD2 induced by the presence of TPL at the domain interface." For such a claim the authors should for example analyze the MD trajectories of residues at the XPB/p52 interface and show the interface is affected by the presence of TPL.

Minor

Point 4. Some labels from Figure 3 are seem confusing and/or are not clearly documented in the Figure legend. In panel (a), was RNA Pol II (as mentioned in the text) or phosphorylated RNA Pol II detected? What does TF in panels (c) and (e) mean?

Point 5. The authors discuss stably paused promoters but do provide a clear definition. Is that proposed by Chen and colleagues (Chen, Genes and Dev, 2014)?

Point 6. The discussion of RNA-Seq and Chip-Seq data for individual genes (mainly in Figure 4d) qualitative and not easy to follow. The authors could annotate the panels from Figure 4d and, for example could indicate the amounts of paused and initiating polymerase.

Author's Response to Decision Letter for (RSOB-20-0050.R0)

See Appendix A.

RSOB-20-0050.R1 (Revision)

Review form: Reviewer 1

Recommendation

Accept as is

Do you have any ethical concerns with this paper?

No

Comments to the Author

Since the authors have addressed all of my concerns, I have no more questions.

Review form: Reviewer 2

Recommendation

Accept as is

Are each of the following suitable for general readers?

- a) **Title**
Yes
- b) **Summary**
Yes
- c) **Introduction**
Yes

Is the supplementary material necessary; and if so is it adequate and clear?

Yes

Do you have any ethical concerns with this paper?

No

Comments to the Author

I'm satisfied by the revised version of the manuscript and have nothing to add. A nice piece of work.

Best regards

Decision letter (RSOB-20-0050.R1)

14-May-2020

Dear Dr Zurita

We are pleased to inform you that your manuscript entitled "Disruption of TFIID activities generates a stress gene expression response and reveals possible new targets against cancer" has been accepted by the Editor for publication in Open Biology.

Please find the referee comments below. No further changes are recommended.

Sincerely,
The Open Biology Team
mailto: openbiology@royalsociety.org

Reviewer(s)' Comments to Author:

Referee: 1

Comments to the Author(s)

Since the authors have addressed all of my concerns, I have no more questions.

Referee: 2

Comments to the Author(s)

I'm satisfied by the revised version of the manuscript and have nothing to add. A nice piece of work.

Best regards

Appendix A Response to the Reviewers

Referee #1:

Uriostegui-Arcos M, et al analyzed the cytotoxic effects of TPL and THZ1 in a MCF10A-ErSrc cell model, revealing that tumor cells are more prone to be targeted by the small molecular inhibitors to TFIID complex. The authors also proved that TPL specifically interferes with the XPB and p52 in TFIID complex, resulting in the reduction of many transcripts by RNAseq and RNAPII-Chip-seq assays. Surprisingly, the authors also reported the RNAPII promoter occupancy of a significant number of genes involved in tumor growth and metastasis did not reduce upon TPL treatment. Such induction of certain genes by TPL can be validated in THZ1 treatment. The depletion of some of these genes can enhance the cytotoxic effects of TPL. The authors made great efforts to dissect the mechanism underlying the cytotoxic effects of TPL and found an interesting potential drug resistant genes. The data are novel and interesting. Below are some concerns

Majors:

- 1) **Fig 1A, the author can use Chou-Talalay method for drug combination, which is the best way to demonstrate whether two drugs possess additive or synergistic effects against certain cancer cells.**

Author reply:

We appreciate this suggestion by the reviewer. We performed the Chou-Talalay method as suggested and it shows that in the Tamoxifen treated cells (transformed cells) a synergic effect is observed with the TPL-THZ1 treatment in all concentrations analysed. Interestingly, in the non-treated cells, a synergic effect is observed only at high concentrations of both drugs. This new interpretation of these results has now been included in the abstract as well as in the results (electronic supplementary material, figure S1b), discussion, and materials and methods sections.

- 2) **The authors found that the induction of several genes involved in tumor growth and metastasis increases upon the treatment of (125 nM) TPL. When looking into closely, such induced genes are early-response genes to TPL treatment, and the induction gradually fade away as shown in Fig 5f, with some genes even lower than DMSO treatment group. It may be due to the cell death which can be inferred from Fig 1. Hence, it is better to check whether such induction is dose-dependent and low dose of TPL can increase such gene expression within 24 h treatment when the majority of cells are still viable.**

Author reply:

This is an important point raised by the reviewer. Her/his observation is correct since the decline in the protein products decrease after 8 h of incubation with TPL, due that most of the cells are dying, and now this is indicated in the results section as follows.

Page 12, Line 7:

“Intriguingly, at 8 h after TPL treatment, the protein levels decline, due that most of the cells start to die”.

About the treatment of the cells at lower doses of TPL for longer times, the experiments are hard to be interpreted, this is due that TPL is not very permeable to the cells, and then the response after incubation with lower doses does not have a homogeneous effect on the cell population. However, we have evaluated the increase of specific RNAs by RT-PCR as response to TPL at 125 nM, besides the kinetic analysis of the protein product, at different times. In general, we observed that, after 1 h of incubation, the RNA increases if compared with the cells with DMSO. Interestingly, the highest accumulation of the different transcripts is reached after 4 h of treatment, which were the conditions used in the RNA-seq and ChIP-seq experiments. We include here a figure of some examples for the reviewer.

- 3) To exclude potential off-target effects of chemicals, it is needed to check whether such genes also change in the XPB knockdown Tam treated or non-treated MCF10A-ErSrc cells.

Author reply:

This is an interesting suggestion to see if TPL is having an additional mechanism that could cause the over-expression of specific genes. Unfortunately, at this point, we are not able to perform this experiment since we do not have a dsRNA against XPB and the conditions in this moment in the world to acquire chemicals from overseas will take an undefined time to perform this complementary experiment. However, since the treatment with THZ1, which inhibits CDK7, also induce the over-expression of several genes that also are activated by TPL, supports that the response is due to the inhibition of transcription. Furthermore, the fact that we observe a very high overexpression of the gene that encodes the large subunit of the RNA Polymerase II (RNAPII), which levels are reduced when transcription is inhibit, suggest a feedback mechanism to maintain

the RNAPII levels supporting that the cell is having a direct response to the inhibition of the TFIID activities.

Minors:

- 1) Although the authors made a clear explanation for “T cells”, an abbreviation for “tamoxifen Treated cells”. There is some confusion because the word “T cells” has its own special meaning, commonly referred to an immune cell population. Therefore, I recommend the authors to change another word for “tamoxifen treated cells”.**

Author reply:

We thank the reviewer for this observation. In the revised manuscript, we now called “TAM cells” to the tamoxifen-treated cells.

- 2) To change P52 and P8 to p52 and p8.**

Author reply:

We changed the nomenclature in the revised manuscript. We have corrected this, thank you.

- 3) Fig 2C, no statistical data on the effects of TPL.**

Author reply:

The requested statistical data have been performed and we included the details in the figure and figure legend.

- 4) In the M&M section, please describe the serum used to culture MCF10A-ERSRC cells.**

Author reply:

In the revised manuscript, we described the serum used as follows:

Materials and Methods, section 4.1, Page. 18, Line: 11:

“...DMEM/F12 medium supplemented with: 5% donor horse serum (charcoal stripping fetal bovine serum was prepared following Struhl’s group protocol for the MCF10A-Er-Src cell line [https://www.encodeproject.org/documents/ae279a0c-4d69-4d46-afa1-33d9657c0ea2/@@download/attachment/MCF-10A_Struhl_protocol.pdf], while the heat inactivated serum was employed for the MCF10A line), 20 ng/mL epidermal growth factor (EGF), 10 µg/mL insulin, 100 µg/mL hydrocortisone, 1 ng/mL cholera toxin, 1 % pen/step, with the addition of puromycin (only in the line MCF10A-Er-Src) [28,60].”

- 5) Fig S1, the analysis of the Annexin V/PI staining is not precise, since the 2nd quadrant for DMSO treatment group is not correct. The author need to use no staining, PI/Annexin VI single staining to divide four quadrants.**

Author reply:

We thank the reviewer for this observation. We have attached the dot plot of the non-stained cells, and the controls for Annexin V +, Viability Dye +, and the double positive Annexin V + / Viability Dye +. We also readjusted the quadrants with Flowjo’s “Curly Quads” option for better

visualize of the distribution the behavior. It is noteworthy that this change does not alter the original interpretation of the data nor the reported cell death distribution.

The updated figure shows that the profile of the unstained cells agrees with the profile of the group of cells treated with DMSO. Note that what appeared to be a poor quadrant adjustment actually corresponds to a population shift entering early apoptosis (Annexin V +), with double positives (Annexin V + / Viability Dye +) being the population corresponding to late apoptosis. However, as we originally proposed, the total death percentages that we consider for the statistic include both quadrants.

Below we show [1] the original figure including the controls requested by the referee and [2] the final figure with readjusting the quadrants.

6) Fig 7, The author described that TPL treatment for 4 h and cells were collected to check knockdown effect. As for the cell survival assay, please indicate the duration for the TPL treatment.

Author reply:

The duration of the TPL treatment after the RNAi transfection is of 4 h and then cell viability was measure. In the bar plot in figure 7, the % of cell survival only shows the cells 72 h after transfection after that the cells were incubated for 4h with TPL. This is indicated in the corresponding results section and in the figure 7 legend as follows:

Results section: 2.6 Depletion of transcripts encoded by genes overexpressed in response to transcriptional stress augments the cells sensitivity to triptolide. Page: 13, Line 25:

“All targeted RNAs were effectively silenced by the corresponding siRNAs, with the maximum reduction at 72 h post-transfection, after that, the cells were incubated with 125 nM of TPL for 4 h and evaluated by immunoblot and flow cytometry”.

Figure7 legend:

“ID2, CRY2, HEXIM1, EPAS1, and LOC730101 were depleted by incubating the corresponding siRNAs at different times. After that, the cells were incubated with 125 nM of TPL for 4 h”.

Referee #2:

The manuscript by Uriostegui-Arcos and colleagues (RSOB-20-0050) reports a detailed analysis of the gene expression responses of an oncogenesis cellular model to TPL (Triptolide) and THZ1 (phenylaminopyrimidine) two well characterized inhibitors of the transcription/DNA repair factor TFIIH.

As expected tumour cells exhibited increased sensitivity to TPL or THZ1 but more importantly, transcriptome analysis combined with chromatin immunoprecipitation demonstrated that although the levels of many transcripts are reduced in TPL-treated cells, a significant number of transcripts increased following TPL treatment, with maintained or increased RNA Pol II-promoter occupancy. These results suggest that cells generate a gene expression response to the stress caused by the inhibition of transcription and reveal possible new targets against cancer. Interestingly, depletion of transcripts encoded by some genes overexpressed in response to transcriptional stress augments the cells sensitivity to TPL.

Overall, the manuscript provides interesting new insights into the regulation of TFIIH activities and new perspectives on the use of substances that target the basal transcription machinery. I have however a few concerns that could/should be taken into consideration

- 1) Point 1. The authors show that TPL interferes with the XPB-p52 interaction, inducing the degradation of the XPB-p52-p8 submodule of TFIIH. The authors should show that as for CAK subunits and XPD, other subunits of core-TFIIH are not affected. Also as these experiments were performed in triplicate, the repeated experiments could be quantified to assess the statistical significance of the claim.**

Author reply:

This is a very important observation by the reviewer. This is because it is known that XPD subunit of TFIIH is also able to interact with CAK, forming an independent subcomplex. Therefore, it is possible that XPD could be not affected like other “core subunits” by the presence of TPL. In order to answer this question, in this new version we introduced a western blot experiment, using the same samples (protein extracts) of the results presented in figure 2a, but using an antibody against the p62 subunits of the core-subcomplex of TFIIH. As mentioned before, we do not observe a decrease in the corresponding protein levels with the TPL treatment. This result is in agree with our proposal that TPL affects the interaction between XPB and p52, destabilizing the XPB-P52-P8 submodule, with no alterations in the rest of the TFIIH subunits.

As also requested by the reviewer, we quantified the different replicas of the experiments in figure 2a, presented in figure S2a of the revised electronic supplementary material.

- 2) **Point 2. A confocal analysis of the Split-GFP complementation assays between p52 and XPB (Figure 2c) shows that fluorescence is un-homogeneously distributed in the nucleoplasm which is striking as both XPB and p52 are in general evenly distributed. This point should be discussed as the accumulation of the XPB-p52-GFP complex could result from aggregation of non-functional proteins. Is the localization of the GFP fragments GFP10 and GFP11 compatible with the formation of a functional TFIIH complex - The distance between the extremities of p52 and XPB fused to GFP10 and GFP11 should be reasonable - ?**

Author reply:

This is a very interesting observation by the reviewer, and we agree with it. The visualization of “foci or grains” when we perform the GFP complementation is very notorious and, as the reviewer suggest, it is possible that it is forming aggregates, probably due to the overexpression of both recombinant proteins, when XPB-GFP10 and p52-GFP11 interact between them and with the rest of GFP. At this point, we do not know if these recombinant proteins are functional, the way to answer this question is to determine if these protein configuration rescue mutant cells in XPB, p52 or both. In previous works by us using *Drosophila* as model, we have demonstrated the fusions of XPB or p52 with the whole GFP are functional, since these recombinant proteins rescue lethal alleles in both genes. However, even if XPB or p52 fused to GFP regions are not functional or can't interact with the rest of TFIIH, for the purpose of the experiment, that is to demonstrate that TPL interferes with the interaction between XPB and p52, the GFP-complementation assay works satisfactorily and allow us to answer the question.

On the other hand, we agree with the reviewer that it is important to show that the association with GFP does not alter the structure of the TFIIH complex. For this, we included in subsection (c) of Figure 2 a three-dimensional representation of the P52-GFP-XPB-complementation. This clearly shows that the localization of the GFPs is compatible with the formation of the XPB-p52-p8 submodule, with a reasonable distance between fused GFP10 and GFP11 fragments. This is now indicated in the corresponding results section as follows:

Page 7; Line 21:

“A three-dimensional representation of the XPB-GFP-p52 complex shows that the localization of GFP is compatible with the interaction between XPB and p52 (figure 2c)”.

- 3) **Point 3. The authors present molecular dynamic simulation indicating that the presence of TPL in the active site of XPB alters the HD1-HD2 interface which suggests that TPL induces a conformational change leading to the destabilization of the XPB/p52 interface. Although this presentation in the result section is in my opinion valid, the authors should take care to avoid over-interpretations. I would not claim, as written in the discussion, that the MD simulation “allowed us to propose that the dissociation/degradation of the XPB-P52-P8 submodule is mainly caused by the separation of XPB HD1-HD2 induced by the presence of TPL at the domain interface.” For such a claim the authors should for example analyze the MD trajectories of residues at the XPB/p52 interface and show the interface is affected by the presence of TPL.**

Author reply:

We thank the reviewer for this observation. To demonstrate this statement, we performed a contact analysis between XPB's HD1 and HD2 domains (in electronic supplementary material,

figure S3d). That analysis showed a decrease in the number of contacts between the domains in the presence of TPL. In this new manuscript, we included per-residue contact analysis between XPB domains, in which the reduction in contacts due to the presence of the inhibitor can be observed (in electronic supplementary material, figure S3g). This analysis confirms that TPL affects the interaction between HD1 and HD2 and could be promoting their separation. This is now indicated in the text in the corresponding results section as follows.

Page: 7; Line: 7:

“During the MD simulations, a per-residue contact analysis shows that the presence of TPL at the HD1-HD2 interface altered the number of contacts between both domains (electronic supplementary material, figure S3d and S3g), which may lead the separation of the domains and that the dissociation could be due to allosteric modulation guided by the loss of interactions between the XPB N-terminal domain (NTD) and p52 and between XPB HD2 and p52/p8.”

Minors

- a) **Point 4. Some labels from Figure 3 are seem confusing and/or are not clearly documented in the Figure legend. In panel (a), was RNA Pol II (as mentioned in the text) or phosphorylated RNA Pol II detected? What does TF in panels (c) and (e) mean?**

Author reply:

In figure 3a, we used an antibody that recognize Ser-5-PO₄ of the RNA Pol II CTD, this is indicated in the figure legend. In this new version, we have changed the term treated cells (T cells), by tamoxifen treated cells (TAM), this is defined at the beginning of the results section as well as in all figure legends.

- b) **Point 5. The authors discuss stably paused promoters but do provide a clear definition. Is that proposed by Chen and colleagues (Chen, Genes and Dev, 2014)?**

Author reply:

Yes, we used the term of stably paused promoters suggested by (Chen et al., 2015). This is now clarified as follows in the results section:

Page: 10; Line: 10:

“Also consistent with the existence of highly stable paused RNAPII at some promoters, maintaining the RNAPII in a paused position even after transcription inhibition by TPL (Chen et al., 2015), since promoters corresponding to paused RNAPII were still detected (figure 4a)”.

- c) **Point 6. The discussion of RNA-Seq and Chip-Seq data for individual genes (mainly in Figure 4d) qualitative and not easy to follow. The authors could annotate the panels from Figure 4d and, for example could indicate the amounts of paused and initiating polymerase.**

Author reply:

We agree with the reviewer’s comment regarding the clarity of this information. Panels show in figure 4d are examples of genes that have different responses to TPL, both the occupancy of the

RNAPII in the promoters and in the body of the genes as well as the amount of transcripts of each gene obtained by the RNA-seq. Following the reviewer's recommendation, we have modified the arrangement of the images to make it clear.

In order to quantify the amount of RNAPII at the promoters in the initiating and in the paused position, the ChIP-seq results are not optimal. For that it is needed to perform ChIP-exo or ChIP-NEXUS that resolve the occupancy of the RNAPII at the level of a single nucleotide (Shao and Zeitlinger, 2017). However, for the questions and results that we have obtained, ChIP-seq experiments were very useful and complement other results.

References

- Chen, F., Gao, X., Shilatifard, A., Shilatifard, A., 2015. Stably paused genes revealed through inhibition of transcription initiation by the TFIIH inhibitor triptolide. *Genes Dev.* 29, 39–47. <https://doi.org/10.1101/gad.246173.114>
- Shao, W., Zeitlinger, J., 2017. Paused RNA polymerase II inhibits new transcriptional initiation. *Nat. Genet.* 49, 1045–1051. <https://doi.org/10.1038/ng.3867>